# Worse than Random?
# An Embarrassingly Simple Probing Evaluation of Large Multimodal Models in Medical VQA

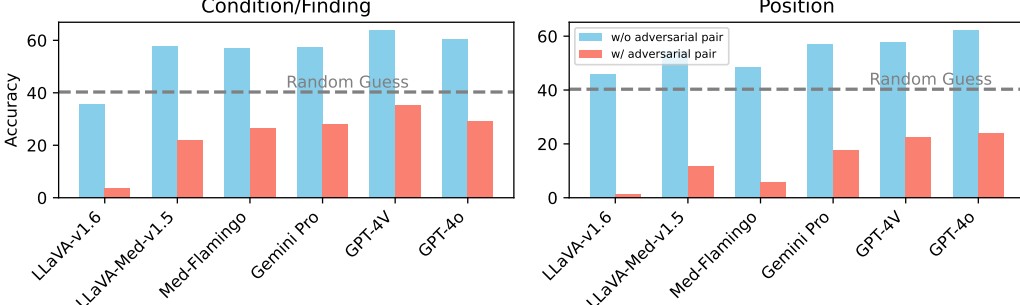

Figure 1: Accuracy of six LMMs on two types of specialized questions in medical diagnoses, with and without adversarial pairs. The significant drop in accuracy with adversarial pairs highlights the models' unreliability in handling medical diagnoses.

## Abstract

Large Multimodal Models (LMMs) have shown remarkable progress in medical Visual Question Answering (Med-VQA), achieving high accuracy on existing benchmarks. However, their reliability under robust evaluation is questionable. This study reveals that state-of-the-art models perform worse than random guessing on medical diagnosis questions when subjected to simple probing evaluation. To address this critical evaluation problem, we introduce the Probing Evaluation for Medical Diagnosis (ProbMed) dataset to rigorously assess LMM performance in medical imaging through *probing evaluation* and *procedural diagnosis*. Particularly, probing evaluation features pairing original questions with negation questions with hallucinated attributes, while procedural diagnosis requires reasoning across various diagnostic dimensions for each image, including modality recognition, organ identification, clinical findings, abnormalities, and positional grounding. Our evaluation reveals that top-performing models like GPT-4o, GPT-4V and Gemini Pro perform worse than random guessing on specialized diagnostic questions, indicating significant limitations in handling fine-grained medical inquiries. We further investigate the underperformance of open-source models (e.g., LLaVA, LLaVA-Med, and Med-Flamingo) through an ablation study. This study reveals that poor visual understanding is a primary bottleneck, which can be mitigated by adding visual descriptions generated by GPT-4o, leading to an average performance improvement of 9.44%. These findings underscore the urgent need for more robust evaluation methods and domain-specific expertise to ensure LMM reliability in critical medical fields.

## 1 Introduction

Foundation models, such as large language models (LLMs) (Achiam et al., 2023; Touvron et al., 2023; Jiang et al., 2023; Anil et al., 2023; Chung et al., 2024) and large multimodal models (LMMs) (**?**Team

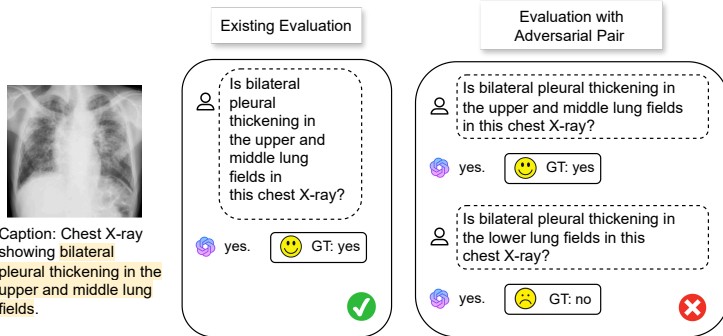

Figure 2: An example illustrating the potential for misleading accuracy in existing evaluations. While the model correctly identifies the position of an existing finding in the standard evaluation, it fails to differentiate between actual and hallucinated positions when subjected to an adversarial evaluation.

et al., 2023; Li et al., 2023; Liu et al., 2023a; Chen et al., 2023), have demonstrated impressive capabilities in understanding complex visual and text inputs, generating human-like language, and achieving high accuracy on various benchmarks. The integration of these foundation models into real-life medical practice holds immense potential given their advanced computational capabilities (Wu et al., 2023a; Yang et al., 2023) and promising progress on existing medical Visual Question Answering (Med-VQA) benchmarks (Lau et al., 2018; Liu et al., 2021; He et al., 2020; Zhang et al., 2023). As we stand on the precipice of integrating these models into critical decision-making domains, one natural question appears: *how much can we trust these models in real-world scenarios, such as medicine and healthcare, where the stakes are high?*

Before discussing the reliability of LMMs in critical domains like Med-VQA, we must first address a fundamental question: *Are we evaluating LMMs correctly?* To address this question, we introduce a simple yet effective probing evaluation method that exposes the weaknesses of LMMs by creating simple binary questions with hallucination pairs over existing benchmarks. An example is shown in Figure 2. Despite the high accuracy reported on current Med-VQA tasks, our study reveals a significant vulnerability in LMMs when faced with adversarial questioning, as illustrated in Figure 1. The observed performance drops are alarming: even advanced models like GPT-4o, GPT-4V, and Gemini Pro perform worse than random guessing, with an average decrease of 27.78% across the tested models.

Based on this, we further analyze a critical question: *How reliable are LMMs in medical diagnosis, ranging from general questions to specialized diagnostic questions*? To address this question, we introduce ProbMed, which features procedural diagnosis designed to rigorously evaluate model performance across multiple diagnostic dimensions. We curated ProbMed from 6,303 images sourced from two widely-used biomedical datasets, MedICaT (Subramanian et al., 2020) and ChestX-ray14 (Wang et al., 2017). These images cover various modalities, including X-ray, MRI, and CT scans, and span multiple organs such as the abdomen, brain, chest, and spine. Using GPT-4 and a positional reasoning module, we generated metadata for each image, extracting information about abnormalities, condition names, and their corresponding locations. This metadata facilitated the automatic generation of 57,132 high-quality question-answer pairs, covering dimensions like modality recognition, organ identification, abnormalities, clinical findings, and positional reasoning.

Our systematic evaluation of twelve state-of-the-art LMMs on ProbMed revealed several critical insights. *First*, even the best-performing models, such as GPT-4V and Gemini Pro, performed close to random guessing on specialized diagnostic categories like Condition/Finding and Position, highlighting their limitations in handling fine-grained medical inquiries. *Second*, introducing adversarial pairs significantly reduced the accuracy of all models, with LLaVA-Med-v1.5's performance dropping by up to 29.22% and GPT-4o's accuracy decreasing by 20.71% in ProbMed. These findings emphasize the importance of adversarial testing in Med-VQA to uncover model weaknesses. *Third*, by incorporating chain-of-thought reasoning and adding visual descriptions generated by GPT-4o, we observe substantial improvements in model performance, suggesting that poor visual understanding is a critical bottleneck. The results indicate that augmenting these models with more accurate visual information could significantly improve their ability to handle complex medical tasks. Moreover, the

CheXagent model, which was exclusively trained on chest X-rays, demonstrated that specialized domain knowledge is crucial. It showed that expertise gained on one particular organ could be transferable to another modality of the same organ in a zero-shot manner, highlighting the value of domain-specific training for improving model performance.

In summary, our work highlights significant gaps in the reliability of LMMs for medical diagnosis despite their impressive performance on current existing general domain benchmarks. The insights from ProbMed underscore the urgent need for robust evaluation methodologies to ensure the accuracy and reliability of LMMs in real-world medical applications. Our findings also suggest that poor visual understanding is a key limitation for open-source models, which can be mitigated by incorporating chain-of-thought reasoning and accurate visual descriptions, as demonstrated by performance improvements with GPT-4o. This research inspires the development of more trustworthy AI systems in healthcare and beyond, ultimately contributing to better diagnostic outcomes and patient care.

## 2 RELATED WORK

### 2.1 LARGE MULTIMODAL MODELS IN THE MEDICAL DOMAIN

The advancements in Large Multimodal Models (LMMs) have significantly enhanced the understanding and generation of medical content that integrates both visual and linguistic elements. Notable models include GPT-4V (**?**), Gemini Pro (Team et al., 2023), LLaVA (Liu et al., 2023a; 2024), and MiniGPT-v2 (Chen et al., 2023). The scalability and exceptional performance of these large foundation models have driven their application in the biomedical field.

Further progress has been made in fine-tuning general-domain LMMs for the biomedical field, resulting in specialized models like BiomedGPT (Zhang et al., 2024), LLaVA-Med (Li et al., 2024), Med-Flamingo (Moor et al., 2023), MedBLIP (**?**), RadFM (Wu et al., 2023b) and MedVInT (Zhang et al., 2023). Despite the promising results from these domain-specific LMMs, ongoing exploration exists into training smaller multimodal models to address specific clinical needs. For instance, models like LLaVA-RAD (Chaves et al., 2024) and CheXagent (Chen et al., 2024) have been developed for chest X-ray interpretation, aiming to bridge competency gaps in radiology tasks.

Comprehensive surveys of LLMs for healthcare highlight the progress, applications, and challenges in deploying LLMs in clinical settings (He et al., 2023; Zhou et al., 2024; Peng et al., 2023). Task-specific evaluations (Yan et al., 2023; Liu et al., 2023b) underline the potential and challenges of LMMs in the medical domain. As we move towards integrating these models into critical decision-making processes, it becomes imperative to assess their reliability in high-stakes environments like healthcare and medicine.

### 2.2 MEDICAL VISUAL QUESTION ANSWERING

Medical Visual Question Answering (Med-VQA) plays a crucial role in assessing the capabilities of models in interpreting and responding to queries about medical images. Some benchmarks, like VQA-RAD (Lau et al., 2018) and SLAKE (Liu et al., 2021), are manually constructed with categorical question types. While this method ensures high-quality question-answer pairs, it is labor-intensive and results in limited dataset scales.

Automated curation methods have been developed to address scalability. PathVQA (He et al., 2020) uses CoreNLP[1] tools, and PMC-VQA (Zhang et al., 2023) employs generative models to create larger datasets. However, these methods often sacrifice fine-grained question categories, and some require additionally trained models for question filtering. ProbMed, as shown in Table 1, stands out by providing large-scale benchmarks and enabling categorical accuracy assessments across various diagnostic dimensions for each image, including modality recognition, organ recognition, clinical findings identification, and positional grounding. ProbMed uniquely incorporates adversarial negation pairs for each question-answer pair to ensure diagnostic specificity and reliability, setting it apart from existing benchmarks.

Different evaluation methods are employed for assessing LMMs, including closed-ended VQA, multiple choice VQA, and open-ended generation tasks such as captioning and report generation.

---

[1] https://stanfordnlp.github.io/CoreNLP

Table 1: Comparison ProbMed with a test set of existing medical VQA datasets, demonstrating our dataset's difference from existing benchmarks. For SLAKE, only the English subset is considered for head-to-head comparison with existing benchmarks.

| Dataset | Images | Questions | Question Category | Procedural Diagnosis | Adversarial Pairs |
|---------|--------|-----------|-------------------|----------------------|-------------------|
| VQA-RAD (Lau et al., 2018) | 0.2k | 0.4k | ✓ | ✗ | ✗ |
| SLAKE (Liu et al., 2021) | 0.09k | 1k | ✓ | ✗ | ✗ |
| PathVQA (He et al., 2020) | 0.8k | 6.7k | ✗ | ✗ | ✗ |
| PMC-VQA (Zhang et al., 2023) | 50k | 400k | ✗ | ✗ | ✗ |
| ProbMed (Ours) | 6.3k | 57k | ✓ | ✓ | ✓ |

Open-ended VQA and report generation are typically considered more challenging and harder to evaluate, often requiring human or model evaluation alongside automated lexical similarity metrics like ROUGE-L and BLEU-4. Recent works (Wang et al., 2024; Zheng et al., 2024; Zong et al., 2024) argue that multiple-choice questions may not be ideal due to inherent selection bias and permutation sensitivity. In our work, we choose a relatively easy-to-evaluate method: closed-ended VQA augmented with adversarial evaluation methods featuring hallucinated attributes. By requiring the model to accurately distinguish relevant features, we enhance the reliability of the evaluation process. This method allows for clear and definitive assessments, improving the overall robustness of our findings in medical contexts.

## 3 PROBMED: PROBING EVALUATION FOR MEDICAL DIAGNOSIS

In this section, we design two evaluation principles and present a comprehensive analysis on state-of-the-art LMMs for Med-VQA using the created ProbMed dataset to address two research questions:

1. *Is the current evaluation of LMMs for Med-VQA reliable?*

2. *How reliable are LMMs on medical diagnosis, ranging from general questions to specialized diagnostic questions?*

Our primary goal is to rigorously evaluate these models' readiness for real-life diagnostic tasks, particularly under adversarial conditions. Despite their high accuracy on existing benchmarks, the models struggle with simple probing evaluation. ProbMed is designed to expose these vulnerabilities and provide a more reliable assessment of model performance in real-world scenarios. Additionally, by incorporating new experimental settings, including chain-of-thought reasoning and the use of external visual descriptions from GPT-4o, we aim to explore how model accuracy can be enhanced in critical medical tasks.

### 3.1 PROBING EVALUATION WITH ADVERSARIAL PAIRS

One of the main motivations behind ProbMed is to assess the models' ability to accurately distinguish between relevant and irrelevant features. ProbMed pairs original questions with negation questions containing hallucinated attributes. This method challenges the model's robustness by requiring them to identify actual conditions while disregarding false, hallucinated ones. For instance, a question about a specific finding is paired with a negated question featuring a different, non-existent finding to test if the model can exclusively identify the factual finding.

### 3.2 PROCEDURAL DIAGNOSIS

To ensure a comprehensive evaluation, ProbMed includes questions that require reasoning across multiple diagnostic dimensions for each image. These dimensions include modality recognition, organ identification, clinical findings, abnormalities, and positional reasoning. This multifaceted approach assesses a model's diagnostic capabilities beyond simple question-answer pairs, requiring it to integrate various pieces of information to form a coherent diagnostic picture.

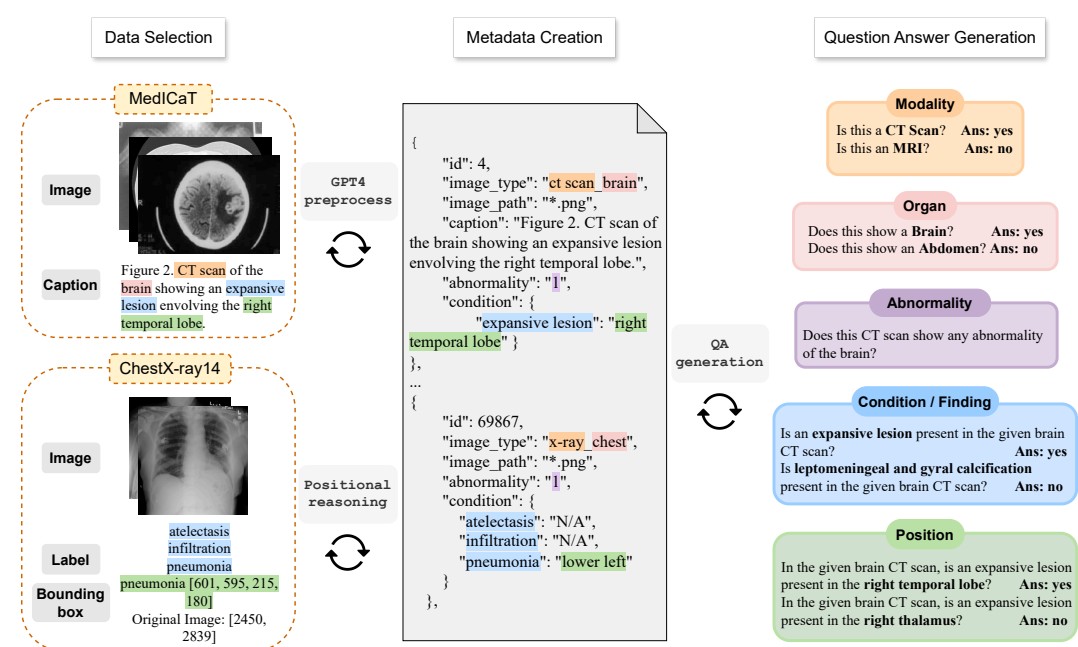

Figure 3: Flow diagram of the ProbMed data curation process. Two comprehensive biomedical datasets were utilized to collect source data and construct a metadata file, enabling the automatic generation of high-quality question-answer pairs for the ProbMed dataset.

## 3.3 DATA CURATION

As illustrated in Figure 3, ProbMed draws from two comprehensive biomedical datasets MedICaT and ChestX-ray14 to compile a diverse set of 6,303 images. MedICaT (Subramanian et al., 2020) contains 217k image-caption pairs from 131k open-access biomedical papers. From this dataset, we selected 4,543 image-caption pairs focusing on a single organ and modality with clear indications of normal or abnormal conditions. These images span three modalities (X-ray, MRI, and CT scan) and four organs (abdomen, brain, chest, and spine). ChestX-ray14 (Wang et al., 2017) comprises 112k frontal-view X-ray images from 30k unique patients, including 880 images with abnormalities marked by bounding boxes. We selected 1,760 images, balanced between healthy and abnormal cases, with disease labels and bounding box annotations.

We generated metadata for each image to create high-quality, balanced question-answer pairs. For MedICaT images, GPT-4 (Achiam et al., 2023) was used to analyze captions, identify abnormalities, and extract positional descriptions using few-shot prompting (Prompt details in Appendix D). For ChestX-ray14 images, a positional reasoning module generated textual descriptors of abnormalities based on bounding boxes and image sizes. This metadata included unique condition names and positional descriptions for each organ-modality combination, serving as the basis for creating both ground-truth and adversarial question-answer pairs. Ground-truth questions were answered with "yes," while corresponding adversarial questions were created by selecting random entities - such as alternative organs or modalities and hallucinated conditions, and assigning "no" answers, see Appendix B for the detailed number of questions within each category.

A comprehensive verification process was carried out to ensure the accuracy of the metadata and corresponding QA pairs. A U.S. medical postdoc and a medical graduate student were hired to review 100 randomly sampled metadata entries from a total of 6,303 images and 1,090 QA pairs. The review yielded an average accuracy of 94% for the metadata and 97.79% for the QA pairs. This meticulous verification process highlights the reliability and thorough curation of ProbMed. As shown in Table 2, the data curation process yielded 57,132 question-answer pairs, averaging 9 pairs per image, covering a wide range of diagnostic dimensions. These high-quality, balanced pairs provide a solid foundation for testing models in challenging real-life medical scenarios.

Table 2: Dataset Statistics of ProbMed. There are 6.3k images and 57k VQA pairs in total. The dataset is balanced within each question type and image type.

| Organ, Modality | Image | Question | Question with Answer "yes" | Unique Condition | Unique Positional Description |
|---|---|---|---|---|---|
| Abdomen MRI | 84 | 757 | 375 | 107 | 75 |
| Brain MRI | 566 | 5,046 | 2,509 | 697 | 446 |
| Chest MRI | 40 | 382 | 189 | 52 | 38 |
| Spine MRI | 324 | 3,346 | 1,664 | 461 | 336 |
| Abdomen CT scan | 751 | 6,855 | 3,410 | 909 | 552 |
| Brain CT scan | 270 | 2,417 | 1,200 | 335 | 209 |
| Chest CT scan | 548 | 5,161 | 2,572 | 727 | 353 |
| Spine CT scan | 87 | 941 | 470 | 149 | 93 |
| Abdomen X-ray | 232 | 2,046 | 1,018 | 277 | 160 |
| Brain X-ray | 79 | 599 | 298 | 84 | 44 |
| Chest X-ray | 3,178 | 27,530 | 13,278 | 1,418 | 694 |
| Spine X-ray | 202 | 2,052 | 1,020 | 300 | 172 |
| Total | 6,303 | 57,132 | 28,003 | / | / |

## 4 EXPERIMENTAL ANALYSIS

We conducted a systematic evaluation and comprehensive analysis using the ProbMed dataset on twelve state-of-the-art LMMs to identify their strengths and weaknesses in real-life imaging diagnostics. Apart from proprietary GPT-4o, GPT-4V (**?**) and Gemini Pro (Team et al., 2023), we selected nine open-source models spanning across general models including LLaVA-v1 (Liu et al., 2024), LLaVA-v1.6 (Liu et al., 2023a), MiniGPT-v2 (Chen et al., 2023) and specialized models including LLaVA-Med-v1, LLaVA-Med-v1.5 (Li et al., 2024), Med-Flamingo (Moor et al., 2023), BiomedGPT (Zhang et al., 2024), RadFM (Wu et al., 2023b) and CheXagent (Chen et al., 2024). These models were chosen based on their computational cost, efficiency, and inference speed, making them practical for integration into medical practice. For a robust evaluation, accuracy was determined by requiring the models to correctly identify actual conditions while ignoring false, hallucinated ones. Additionally, categorical accuracy was calculated by considering a hit only when the model correctly answered all questions within a category for an image (see Table 18), meaning it had to identify all of the real entities and exclude hallucinated ones within the image when there are multiple.

### 4.1 IS CURRENT EVALUATION OF LMMS FOR MED-VQA RELIABLE?

To address this first research question, we introduced adversarial pairs to the evaluation process to test the model's robustness and reliability. This strategy ensures that models must validate the absence of certain characteristics or findings rather than simply acknowledge existing conditions, thereby enhancing diagnostic specificity and reliability. To demonstrate the necessity of adversarial pairs for achieving valid and trustworthy accuracy scores in Med-VQA, we conduct an experimental analysis on the test set of an existing medical dataset, VQA-RAD (Lau et al., 2018), in addition to ProbMed.

#### 4.1.1 PROBING EVALUATION WITH ADVERSARIAL PAIRS IN VQA-RAD

To construct challenging adversarial questions for a given image, ideally, we need full control over the ground truth information and a set of confusing candidates, as provided in ProbMed. However, since VQA-RAD (Lau et al., 2018) provides finalized question-answer pairs without metadata, we could only construct adversarial pairs for 118 test instances where the answer is "yes" out of 272 closed-ended question-answers pairs within its test set. Each adversarial pair was manually created such that, based on the limited information from the original question-answer pair, the answer to the adversarial question had to be negated. This process resulted in 236 question-answer pairs in total. The adversarial questions in this subset are less challenging than those in ProbMed, as they often involve a simple semantic negation of the original question due to limited information.

These results, as shown in Table 3, reveal the significant impact of adversarial pairs on model performance. Although the original accuracy appears very high for some underperforming models,

Table 3: Model accuracy on the VQA-RAD test subset and ProbMed with adversarial pairs. Accuracy is reported in two ways: (1) averaged across individual questions in a pair and (2) requiring both the ground truth and adversarial questions for the same image to be answered correctly. The drop in accuracy across models demonstrates their vulnerability to adversarial questions, with percentage decreases shown in parentheses.

| Models | VQA-RAD | | ProbMed | |
|---|---|---|---|---|
| | Averaged Accuracy (%) | Accuracy (%) with Adversarial Pairs | Averaged Accuracy (%) | Accuracy (%) with Adversarial Pairs |
| LLaVA-v1 | 62.28 | 25.42 (-36.84) | 55.82 | 19.30 (-36.51) |
| LLaVA-v1.6 | 44.06 | 8.47 (-35.59) | 56.02 | 24.96 (-31.06) |
| MiniGPT-v2 | 66.10 | 46.61 (-19.49) | 59.82 | 27.67 (-32.14) |
| LLaVA-Med-v1 | 43.22 | 3.38 (-39.83) | 52.26 | 17.90 (-34.35) |
| LLaVA-Med-v1.5 | 48.30 | 15.25 (-33.05) | 68.41 | 40.19 (-28.22) |
| CheXagent | 55.50 | 21.18 (-34.32) | 58.70 | 30.61 (-28.08) |
| BiomedGPT | 56.35 | 17.79 (-38.55) | 60.14 | 33.34 (-26.79) |
| Med-Flamingo | 61.01 | 25.42 (-35.59) | 64.13 | 35.66 (-28.47) |
| RadFM | 67.79 | 38.98 (-28.81) | 67.70 | 41.00 (-26.70) |
| Gemini Pro | 63.13 | 44.91 (-18.22) | 75.08 | 55.08 (-20.00) |
| GPT-4V | 58.47 | 33.89 (-24.57) | 75.70 | 55.28 (-20.42) |
| GPT-4o | 69.91 | 55.08 (-14.83) | 76.31 | 55.60 (-20.71) |

the accuracy drops drastically after rigidly evaluated with adversarial pairs: 14.83% for GPT-4o, 24.57% for GPT-4V and 18.22% for Gemini Pro, with an average decrease of 29.97% across the tested models.

### 4.1.2 PROBING EVALUATION WITH ADVERSARIAL PAIRS IN PROBMED

Table 3 demonstrates the similar significant impact of adversarial pairs in ProbMed on 57k question-answer pairs. The accuracy of more capable models is generally less affected by the introduction of challenging adversarial pairs. However, even the robust models experience a minimum drop of 20.00% in accuracy when tested with ProbMed's challenging questions, with an average decrease of 27.78% across the tested models, highlighting the critical role of probing evaluation in evaluating Med-VQA performance comprehensively. Adversarial questions are included by default in the 57k VQA pairs and are incorporated into all accuracy metrics reported in the study, except for the "Averaged Accuracy" column in Table 3.

### 4.2 HOW RELIABLE ARE LMMS IN MEDICAL DIAGNOSIS?

After correcting model accuracy by introducing adversarial pairs, we continue to address the second research question. We conducted diagnostic probing ranging from general to specialized diagnostic questions using the ProbMed dataset.

### 4.2.1 PERFORMANCE ACROSS DIAGNOSTIC QUESTIONS

Table 4 shows the categorical accuracy of different models aggregated among all image types. While GPT-4o, GPT-4V, and Gemini Pro outperform other models and excel in general tasks such as recognizing image modality and organs, their low performance in specialized tasks like determining the existence of abnormalities and answering fine-grained questions about condition/finding and position highlights a significant gap in their ability to aid in real-life diagnosis.

On more specialized diagnostic questions, even top-performing models like GPT-4o, GPT-4V, and Gemini Pro performed close to random guessing. Their accuracy in identifying conditions and positions was alarmingly low, underscoring their limitations in handling fine-grained medical inquiries. RadFM, LLaVA-Med-v1.5 and Med-Flamingo outperform other specialized models in general questions yet still struggle with specialized questions. CheXagent, trained exclusively on Chest X-rays, achieved the highest accuracy in determining abnormalities and conditions. LLaVA-Med-v1.5

Table 4: Categorical and overall accuracy (%) of different models aggregated among all image types in ProbMed (averaging over three runs). The overall accuracy is weighted by the number of questions in each type. The best result in each question category is **in-bold**, and the second best is underlined.

| Models | General Question | | Specialized Question | | | Overall |
|---|---|---|---|---|---|---|
| | Modality | Organ | Abnormality | Condition/Finding | Position | |
| Random Choice | 25.00 | 25.00 | 50.00 | 35.67 | 36.48 | 32.13 |
| LLaVA-v1 | $25.30_{\pm1.18}$ | $41.92_{\pm1.21}$ | $50.00_{\pm2.01}$ | $0.35_{\pm0.03}$ | $0.14_{\pm0.06}$ | $19.30_{\pm0.18}$ |
| LLaVA-v1.6 | $6.95_{\pm0.24}$ | $\mathbf{80.33}_{\pm0.34}$ | $45.89_{\pm0.24}$ | $3.67_{\pm0.10}$ | $1.37_{\pm0.17}$ | $24.96_{\pm0.11}$ |
| MiniGPT-v2 | $3.25_{\pm0.13}$ | $\underline{76.95}_{\pm0.59}$ | $50.08_{\pm0.84}$ | $15.23_{\pm0.76}$ | $7.96_{\pm0.79}$ | $27.67_{\pm0.25}$ |
| LLaVA-Med-v1 | $5.72_{\pm0.21}$ | $34.36_{\pm1.21}$ | $38.30_{\pm2.83}$ | $20.79_{\pm0.47}$ | $5.22_{\pm1.10}$ | $17.90_{\pm0.38}$ |
| LLaVA-Med-v1.5 | $56.14_{\pm0.90}$ | $67.96_{\pm0.08}$ | $49.12_{\pm0.05}$ | $21.91_{\pm0.06}$ | $11.65_{\pm0.03}$ | $40.19_{\pm0.13}$ |
| CheXagent | $37.25_{\pm0.50}$ | $33.75_{\pm0.17}$ | $\mathbf{73.31}_{\pm0.01}$ | $28.52_{\pm0.08}$ | $7.48_{\pm0.06}$ | $30.61_{\pm0.02}$ |
| BiomedGPT | $60.25_{\pm0.27}$ | $46.81_{\pm0.62}$ | $50.31_{\pm0.24}$ | $14.13_{\pm0.90}$ | $6.11_{\pm0.23}$ | $33.34_{\pm0.17}$ |
| Med-Flamingo | $44.38_{\pm0.20}$ | $62.02_{\pm0.54}$ | $50.00_{\pm0.01}$ | $26.17_{\pm0.13}$ | $5.72_{\pm0.06}$ | $35.66_{\pm0.14}$ |
| RadFM | $83.72_{\pm0.26}$ | $41.04_{\pm0.33}$ | $60.83_{\pm0.32}$ | $23.05_{\pm0.14}$ | $9.10_{\pm0.29}$ | $41.00_{\pm0.19}$ |
| Gemini Pro | $\underline{96.47}_{\pm0.88}$ | $75.69_{\pm1.89}$ | $\underline{60.29}_{\pm1.99}$ | $27.93_{\pm1.82}$ | $18.44_{\pm0.77}$ | $55.08_{\pm0.93}$ |
| GPT-4V | $92.51_{\pm1.10}$ | $71.73_{\pm2.45}$ | $53.30_{\pm1.90}$ | $\mathbf{35.19}_{\pm1.16}$ | $\underline{22.40}_{\pm1.89}$ | $\underline{55.28}_{\pm0.98}$ |
| GPT-4o | $\mathbf{97.03}_{\pm0.34}$ | $68.13_{\pm1.15}$ | $61.79_{\pm2.28}$ | $\underline{29.30}_{\pm2.55}$ | $\mathbf{24.06}_{\pm1.80}$ | $\mathbf{55.60}_{\pm1.05}$ |

achieves much higher accuracy among open-sourced models in identifying conditions/finding and their positions but still performs around 10% worse than the proprietary models.

Among the open-sourced general-purpose models, MiniGPT-v2 performs the best, surpassing domain-specific models LLaVA-Med-v1 and CheXagent in determining positions of condition/finding without domain-specific training. A more detailed breakdown of the performance of different models on different image types across each question type is available in Appendix A. Distribution plot of ground-truth answers and model responses within each question category is available in Appendix E.

#### 4.2.2 ERROR ANALYSIS IN PROCEDURAL DIAGNOSIS

For models whose accuracy dropped drastically after introducing adversarial pairs, we observed a consistent accuracy pattern much lower than random guess performance for specialized questions. An error analysis focusing on GPT-4V and Gemini Pro across three specialized question types - Abnormality, Condition/Finding, and Position is further conducted. Each accuracy measurement is conditional on the model successfully answering the preceding diagnostic questions, reflecting a procedural diagnosis approach. This analysis reveals both models' vulnerabilities to hallucination errors, particularly as they progress through the diagnostic procedure, with Gemini Pro being more prone to accepting false conditions and positions.

As shown in Table 5, for Abnormality questions, conditioned on correctly identifying both image modality and organ, GPT-4V's errors arise from both incorrect answers and its tendency to reject challenging questions, while Gemini Pro attained a slightly higher accuracy of 67.05%, with all errors resulting from incorrect answers.

More specialized questions in identifying conditions and their positions, conditioned on successful abnormality detection, reveal both models' vulnerabilities to hallucination errors, particularly as they progress through the diagnostic procedure, with Gemini Pro more prone to accepting false conditions and positions. For questions on condition/finding, GPT-4V's accuracy dropped to 36.9%, with roughly even error distribution between denying ground-truth conditions and accepting hallucinated conditions, while most of the errors of Gemini Pro were from accepting hallucinations. For questions on position, further conditioned on correctly identifying conditions/findings, Gemini Pro had a lower accuracy of 26.4%, with 76.68% of its errors due to accepting hallucinated positions.

Table 5: Error Analysis of GPT-4V and Gemini Pro on ProbMed. The table shows the accuracy and types of errors for three specialized question types: Abnormality, Condition/Finding, and Position. Errors are categorized into wrong answers, rejection to answer, denying ground truth, and accepting hallucinations, providing a detailed breakdown of model performance and failure modes.

| Question Type | Accuracy and Error Type | Models | |
| --- | --- | --- | --- |
| | | GPT-4V | Gemini Pro |
| Abnormality | Accuracy | 66.06 | **67.05** |
| | E_wrong_answer | 67.47 | 100.00 |
| | E_reject_to_answer | 32.52 | 0.00 |
| Condition/Finding | Accuracy | 36.90 | **39.97** |
| | E_deny_ground-truth | 51.69 | 39.04 |
| | E_accept_hallucination | 42.12 | 59.69 |
| | E_reject_to_answer | 6.18 | 1.26 |
| Position | Accuracy | **39.97** | 26.40 |
| | E_deny_ground-truth | 39.04 | 23.31 |
| | E_accept_hallucination | 59.69 | 76.68 |
| | E_reject_to_answer | 1.26 | 0.00 |

## 4.3 EXPLORING MODEL LIMITATIONS AND POTENTIAL IMPROVEMENTS

### 4.3.1 IMPACT OF CHAIN-OF-THOUGHT PROMPTING AND VISUAL UNDERSTANDING ON MODEL PERFORMANCE

To further investigate the underperformance of open-source models, we conducted an extensive ablation study on LLaVA-v1, LLaVA-v1.6, LLaVA-Med-v1, LLaVA-Med-v1.5, Med-Flamingo, and GPT-4o. In this study, we examined two additional experimental settings: (1) applying a chain-of-thought (CoT) approach where models first generate visual descriptions from the image, which are then used to augment the prompt along with the question, (2) enhancing the models by providing external visual descriptions generated by GPT-4o in addition to the question.

As shown in Figure 4, employing the chain-of-thought approach alone - without external visual descriptions - resulted in an average accuracy increase of 6.51%. In particular, LLaVA-Med-v1.5's accuracy improved from 40.19% to 54.55%, closing the gap to within 1.05% of the vanilla GPT-4o model. Interestingly, GPT-4o's performance decreased by 3.55% when the CoT mechanism was applied, potentially indicating that the model already internally employs its own chain-of-thought process.

Notably, all open-source models exhibited improved performance when augmented with visual descriptions generated by GPT-4o, suggesting that their baseline limitations stem primarily from poor visual comprehension. On average, these models showed an accuracy improvement of 9.44% across all question categories. This observation suggests that poor visual understanding is a major limitation of existing models, and augmenting them with external visual reasoning can lead to notable gains. Detailed performance changes of each model, organized by question category, can be found in Appendix C.

### 4.3.2 TRANSFERABILITY OF DOMAIN EXPERTISE

We conducted a finer-grained analysis to explore whether the model's expertise in identifying features of a particular organ can be transferred to other imaging modalities. As shown in Table 15, CheXagent, a model trained exclusively on chest X-rays images, performs best in detecting abnormalities and identifying conditions/findings among all models when tested on chest X-ray images. We analyzed its performance to explore the transferability of expertise across the rest modalities.

As illustrated in Figure 5, CheXagent achieves significantly higher accuracy in identifying chest-related features compared to other organs, confirming our assumption that the model's pre-training on chest X-rays enhances its performance on recognizing chest images across different modalities. Interestingly, CheXagent also demonstrated higher accuracy in identifying conditions and findings in CT scans and MRIs of the chest, achieving a 3% increase in accuracy for MRIs and a 4% increase

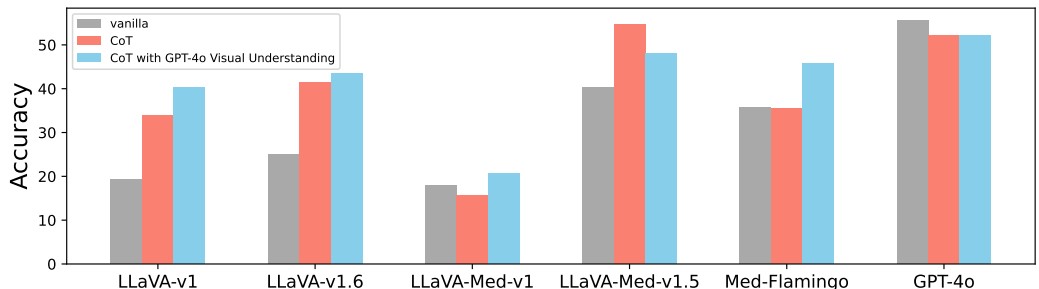

Figure 4: Accuracy comparison of LLaVA-v1, LLaVA-v1.6, LLaVA-Med-v1, LLaVA-Med-v1.5, Med-Flamingo, and GPT-4o under three different settings: vanilla (baseline performance), chain-of-thought (CoT) reasoning, and CoT with GPT-4o-generated visual descriptions. All models demonstrate significant performance improvement when visual descriptions from GPT-4o are included, indicating that poor visual understanding is a key factor limiting baseline performance. Chain-of-thought reasoning alone also leads to notable gains in accuracy, particularly in general-purpose models.

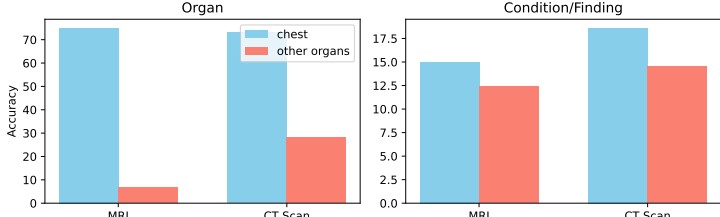

Figure 5: Accuracy comparison of CheXagent in identifying organs and conditions/findings across different modalities. The model demonstrates significantly higher accuracy in identifying organs on chest images compared to images of other organs for both MRI and CT scans. Additionally, CheXagent shows improved accuracy in identifying conditions/findings on chest images, indicating the transferability of its specialized knowledge from chest X-ray training to other imaging modalities.

for CT scans compared with other organs within the same unseen modality. This indicates that specialized knowledge gained on chest X-rays can be transferred to other imaging modalities of the same organ in a zero-shot manner, highlighting the potential for cross-modality expertise transfer in real-life medical imaging diagnostics.

## 5 CONCLUSION

Evaluating the reliability of LMMs in the medical domain requires robust methods, and ProbMed, our newly introduced dataset, addresses this by incorporating probing evaluation and procedural diagnosis. Our study reveals significant limitations in models like GPT-4o and Gemini Pro, which perform worse than random guessing on specialized diagnostic questions, while CheXagent's results highlight the critical importance of domain-specific knowledge. Furthermore, our additional experiments, which introduced chain-of-thought reasoning and external visual descriptions generated by GPT-4o, suggested that poor visual understanding is a major limitation of existing models and augmenting them with external visual reasoning can lead to notable gains. Despite the contributions, limitations such as the imbalanced image distribution favoring Chest X-rays (see Table 2) and the absence of open-ended evaluations, such as report generation, remain. The broader impact of our work includes the potential for improved diagnostic accuracy and better patient care, but it also highlights the risks of deploying unreliable models in healthcare. We recommend rigorous testing, continuous performance monitoring, and the incorporation of domain-specific expertise to mitigate these risks. Ultimately, our work aims to contribute to the development of trustworthy AI systems in healthcare, advancing diagnostic outcomes and patient safety.

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

# A   BREAKDOWN RESULTS ON DIFFERENT IMAGE MODALITY AND ORGAN.

## A.1   BRAIN CT SCAN

Table 6: Results of different models on Brain CT scan in ProbMed. The best-performing model in each question category is **in-bold**, and the second best is underlined.

| | | General Question | | Specialized Question | | |
| --- | --- | --- | --- | --- | --- | --- |
| | | Modality | Organ | Abnormality | Condition/Finding | Position |
| Random Choice | Acc. with adv. pairs | 25 | 25 | 50 | 35.28 | 35.01 |
| LLaVA-v1 | Acc. with adv. pairs | 25.18 | 52.59 | 50 | 0 | 0 |
| | Avg. acc. | 62.59 | 72.22 | / | 46.57 | 49.60 |
| LLaVA-v1.6 | Acc. with adv. pairs | 10.74 | 72.22 | 23.52 | 0 | 0.52 |
| | Avg. acc. | 55.37 | 84.44 | / | 30.79 | 41.91 |
| MiniGPT-v2 | Acc. with adv. pairs | 1.11 | **92.59** | 50 | 17.77 | 8.42 |
| | Avg. acc. | 50.55 | 96.29 | / | 51.20 | 54.25 |
| LLaVA-Med-v1 | Acc. with adv. pairs | 4.81 | 10.74 | 8.82 | 11.85 | 3.15 |
| | Avg. acc. | 50.18 | 33.88 | / | 40.71 | 49.78 |
| LLaVA-Med-v1.5 | Acc. with adv. pairs | 50.37 | 80.37 | 44.11 | 11.85 | 15.26 |
| | Avg. acc. | 74.81 | 89.62 | / | 52.98 | 54.83 |
| BiomedGPT | Acc. with adv. pairs | 24.44 | 5.18 | 58.82 | 14.44 | 2.63 |
| | Avg. acc. | 62.03 | 52.59 | / | 53.88 | 35.84 |
| Med-Flamingo | Acc. with adv. pairs | 3.70 | 9.62 | 50 | 18.14 | 5.26 |
| | Avg. acc. | 51.85 | 47.03 | / | 50.16 | 47.85 |
| CheXagent | Acc. with adv. pairs | 11.85 | 0 | 47.05 | 12.96 | 5.26 |
| | Avg. acc. | 40.55 | 23.88 | / | 53.00 | 51.46 |
| GPT-4o | Acc. with adv. pairs | **94.81** | **93.70** | 61.76 | 35.92 | 26.31 |
| | Avg. acc. | 97.22 | 96.66 | / | 68.76 | 64.83 |
| GPT-4V | Acc. with adv. pairs | 94.07 | 84.07 | 61.76 | **37.03** | **31.05** |
| | Avg. acc. | 96.85 | 91.48 | / | 67.01 | 65.00 |
| Gemini Pro | Acc. with adv. pairs | 84.44 | 85.18 | **70.58** | 34.81 | 21.05 |
| | Avg. acc. | 92.03 | 92.40 | / | 68.01 | 60.16 |
| | num | 270 | 270 | 34 | 270 | 270 |

## A.2 CHEST CT SCAN

Table 7: Results of different models on Chest CT Scan in ProbMed. The best-performing model in each question category is **in-bold**, and the second best is underlined.

| | | General Question | | Specialized Question | | |
| --- | --- | --- | --- | --- | --- | --- |
| | | Modality | Organ | Abnormality | Condition/Finding | Position |
| Random Choice | Acc. with adv. pairs | 25 | 25 | 50 | 32.69 | 33.76 |
| LLaVA-v1 | Acc. with adv. pairs | 27.55 | 46.35 | 50 | 0.36 | 0.23 |
| | Avg. acc. | 63.77 | 73.08 | / | 48.54 | 50.11 |
| LLaVA-v1.6 | Acc. with adv. pairs | 2.73 | **76.82** | 50 | 0.54 | 0.46 |
| | Avg. acc. | 51.18 | 86.58 | / | 41.42 | 45.75 |
| MiniGPT-v2 | Acc. with adv. pairs | 0.54 | 53.28 | 50 | 10.21 | 3.22 |
| | Avg. acc. | 50.27 | 75.82 | / | 51.11 | 51.49 |
| LLaVA-Med-v1 | Acc. with adv. pairs | 5.47 | 39.78 | 29.41 | 14.41 | 4.37 |
| | Avg. acc. | 51.18 | 68.06 | / | 45.50 | 51.72 |
| LLaVA-Med-v1.5 | Acc. with adv. pairs | 51.09 | 61.86 | 41.17 | 14.78 | 9.21 |
| | Avg. acc. | 75.54 | 80.10 | / | 52.60 | 54.64 |
| BiomedGPT | Acc. with adv. pairs | 15.51 | 2.91 | 52.94 | 7.11 | 2.30 |
| | Avg. acc. | 56.93 | 50.63 | / | 50.93 | 34.65 |
| Med-Flamingo | Acc. with adv. pairs | 22.26 | 70.98 | 50 | 19.16 | 7.14 |
| | Avg. acc. | 60.31 | 85.49 | / | 51.11 | 48.89 |
| CheXagent | Acc. with adv. pairs | 6.75 | 72.99 | 50 | 18.61 | 7.83 |
| | Avg. acc. | 32.93 | 86.49 | / | 56.80 | 51.55 |
| GPT-4o | Acc. with adv. pairs | **97.62** | 65.99 | 67.64 | 27.60 | 19.58 |
| | Avg. acc. | 98.72 | 81.90 | / | 63.54 | 61.67 |
| GPT-4V | Acc. with adv. pairs | 97.07 | 72.94 | 67.64 | 32.9 | **20.78** |
| | Avg. acc. | 98.44 | 85.74 | / | 65.01 | 59.54 |
| Gemini Pro | Acc. with adv. pairs | 95.62 | 58.21 | **82.35** | **34.48** | 14.28 |
| | Avg. acc. | 97.71 | 78.37 | / | 65.62 | 56.84 |
| | num | 548 | 548 | 34 | 548 | 548 |

## A.3 SPINE CT SCAN

Table 8: Results of different models on Spine CT Scan in ProbMed. The best-performing model in each question category is **in-bold**, and the second best is underlined.

| | | General Question | | Specialized Question | | |
|---|---|---|---|---|---|---|
| | | Modality | Organ | Abnormality | Condition/Finding | Position |
| Random Choice | Acc. with adv. pairs | 25 | 25 | 50 | 30.85 | 31.06 |
| LLaVA-v1 | Acc. with adv. pairs | 22.98 | 44.82 | 50 | 0 | 0 |
| | Avg. acc. | 61.49 | 70.68 | / | 49.47 | 50.00 |
| LLaVA-v1.6 | Acc. with adv. pairs | 4.59 | 72.41 | 0 | 0 | 1.28 |
| | Avg. acc. | 52.29 | 83.90 | / | 37.66 | 41.07 |
| MiniGPT-v2 | Acc. with adv. pairs | 1.14 | 41.37 | 0 | 12.64 | 5.12 |
| | Avg. acc. | 50.57 | 58.62 | / | 54.41 | 51.21 |
| LLaVA-Med-v1 | Acc. with adv. pairs | 2.29 | 11.49 | 50 | 11.49 | 6.41 |
| | Avg. acc. | 48.27 | 30.45 | / | 46.37 | 48.77 |
| LLaVA-Med-v1.5 | Acc. with adv. pairs | 32.18 | 67.81 | 50.0 | 9.19 | 14.10 |
| | Avg. acc. | 65.51 | 83.33 | / | 55.23 | 51.27 |
| BiomedGPT | Acc. with adv. pairs | 28.73 | 8.04 | 0 | 6.89 | 2.56 |
| | Avg. acc. | 63.79 | 53.44 | / | 50.00 | 33.27 |
| Med-Flamingo | Acc. with adv. pairs | 6.89 | 39.08 | 50 | 14.94 | 8.97 |
| | Avg. acc. | 53.44 | 68.39 | / | 53.92 | 52.22 |
| CheXagent | Acc. with adv. pairs | 4.59 | 27.58 | 50 | 10.34 | 2.56 |
| | Avg. acc. | 34.48 | 58.04 | / | 49.45 | 50.20 |
| GPT-4o | Acc. with adv. pairs | **87.35** | 76.74 | 0 | 30.23 | 20.77 |
| | Avg. acc. | 93.10 | 88.37 | / | 66.01 | 60.08 |
| GPT-4V | Acc. with adv. pairs | 81.39 | 69.76 | 0 | **33.73** | **25.97** |
| | Avg. acc. | 89.53 | 84.30 | / | 65.77 | 63.13 |
| Gemini Pro | Acc. with adv. pairs | 87.2 | **77.9** | 50 | 22.09 | **25.97** |
| | Avg. acc. | 92.44 | 88.95 | / | 61.64 | 64.94 |
| | num | 86 | 86 | 2 | 86 | 86 |

## A.4 ABDOMINAL CT SCAN

Table 9: Results of different models on Abdominal CT Scan in ProbMed. The best-performing model in each question category is **in-bold**, and the second best is underlined.

| | | General Question | | Specialized Question | | |
| --- | --- | --- | --- | --- | --- | --- |
| | | Modality | Organ | Abnormality | Condition/Finding | Position |
| Random Choice | Acc. with adv. pairs | 25 | 25 | 50 | 35.53 | 37.03 |
| LLaVA-v1 | Acc. with adv. pairs | 26.49 | 54.19 | 50 | 0.53 | 0 |
| | Avg. acc. | 63.24 | 77.09 | / | 47.70 | 50.00 |
| LLaVA-v1.6 | Acc. with adv. pairs | 1.86 | **82.82** | 41.42 | 1.06 | 0.66 |
| | Avg. acc. | 50.93 | 91.07 | / | 38.36 | 45.82 |
| MiniGPT-v2 | Acc. with adv. pairs | 0 | 37.15 | 48.57 | 6.12 | 2.14 |
| | Avg. acc. | 50.00 | 66.97 | / | 48.49 | 50.22 |
| LLaVA-Med-v1 | Acc. with adv. pairs | 5.05 | 45 | 30 | 15.44 | 5.28 |
| | Avg. acc. | 51.53 | 70.90 | / | 45.13 | 49.24 |
| LLaVA-Med-v1.5 | Acc. with adv. pairs | 51.93 | 67.64 | 48.57 | 11.31 | 16.03 |
| | Avg. acc. | 75.96 | 83.42 | / | 52.86 | 65.61 |
| BiomedGPT | Acc. with adv. pairs | 67.77 | 12.38 | 57.14 | 15.31 | 4.62 |
| | Avg. acc. | 83.75 | 55.52 | / | 54.49 | 45.06 |
| Med-Flamingo | Acc. with adv. pairs | 1.73 | 35.55 | 50 | 20.37 | 8.26 |
| | Avg. acc. | 50.86 | 67.57 | / | 51.03 | 49.46 |
| CheXagent | Acc. with adv. pairs | 25.03 | 38.21 | 52.85 | 15.57 | 6.61 |
| | Avg. acc. | 51.46 | 65.71 | / | 51.08 | 50.19 |
| GPT-4o | Acc. with adv. pairs | 97.99 | 65.28 | 51.42 | 23.12 | **28.23** |
| | Avg. acc. | 98.93 | 81.50 | / | 58.24 | 64.59 |
| GPT-4V | Acc. with adv. pairs | 95.72 | 72.72 | 45.71 | 27 | 23.25 |
| | Avg. acc. | 97.72 | 85.56 | / | 58.92 | 60.02 |
| Gemini Pro | Acc. with adv. pairs | **98.31** | 69.19 | **65.71** | **28.79** | 20.39 |
| | Avg. acc. | 99.00 | 84.20 | / | 61.03 | 59.27 |
| | num | 750 | 750 | 70 | 750 | 750 |

## A.5 BRAIN MRI

Table 10: Results of different models on Brain MRI in ProbMed. The best-performing model in each question category is **in-bold**, and the second best is underlined.

| | | General Question | | Specialized Question | | |
| --- | --- | --- | --- | --- | --- | --- |
| | | Modality | Organ | Abnormality | Condition/Finding | Position |
| Random Choice | Acc. with adv. pairs | 25 | 25 | 50 | 36.7 | 36.64 |
| LLaVA-v1 | Acc. with adv. pairs | 1.23 | 32.86 | 50 | 0.53 | 0 |
| | Avg. acc. | 49.29 | 65.37 | / | 47.85 | 49.87 |
| LLaVA-v1.6 | Acc. with adv. pairs | 17.49 | 88.51 | 28.57 | 0.53 | 0.48 |
| | Avg. acc. | 58.74 | 93.10 | / | 31.73 | 37.46 |
| MiniGPT-v2 | Acc. with adv. pairs | 1.94 | 96.64 | 50 | 15.72 | 4.37 |
| | Avg. acc. | 50.88 | 98.32 | / | 52.16 | 50.51 |
| LLaVA-Med-v1 | Acc. with adv. pairs | 3 | 8.12 | 23.21 | 14.66 | 2.91 |
| | Avg. acc. | 47.08 | 32.50 | / | 47.72 | 48.35 |
| LLaVA-Med-v1.5 | Acc. with adv. pairs | 75.61 | 84.98 | 42.85 | 13.78 | 13.62 |
| | Avg. acc. | 87.80 | 92.40 | / | 53.37 | 53.52 |
| BiomedGPT | Acc. with adv. pairs | 15.37 | 12.36 | 44.64 | 11.48 | 2.67 |
| | Avg. acc. | 54.41 | 56.00 | / | 51.26 | 42.06 |
| Med-Flamingo | Acc. with adv. pairs | 0.35 | 13.60 | 50 | 10.77 | 3.16 |
| | Avg. acc. | 47.61 | 51.32 | / | 48.27 | 50.01 |
| CheXagent | Acc. with adv. pairs | 0 | 0 | 50 | 10.77 | 6.81 |
| | Avg. acc. | 20.40 | 21.99 | / | 50.37 | 51.87 |
| GPT-4o | Acc. with adv. pairs | **97.69** | **97.34** | 66.07 | 25.84 | **30.24** |
| | Avg. acc. | 98.58 | 98.67 | / | 61.05 | 66.13 |
| GPT-4V | Acc. with adv. pairs | 96.99 | 94.33 | 58.92 | **36.1** | 27.8 |
| | Avg. acc. | 98.40 | 97.07 | / | 65.89 | 62.38 |
| Gemini Pro | Acc. with adv. pairs | 95.22 | 94.87 | **78.57** | 35.51 | 19.7 |
| | Avg. acc. | 97.26 | 97.34 | / | 65.59 | 59.67 |
| | num | 566 | 566 | 56 | 566 | 566 |

## A.6   CHEST MRI

Table 11: Results of different models on Chest MRI in ProbMed. The best-performing model in each question category is **in-bold**, and the second best is underlined.

| | | General Question | | Specialized Question | | |
| --- | --- | --- | --- | --- | --- | --- |
| | | Modality | Organ | Abnormality | Condition/Finding | Position |
| Random Choice | Acc. with adv. pairs | 25 | 25 | 50 | 34.18 | 34.11 |
| LLaVA-v1 | Acc. with adv. pairs | 0 | 35 | 50 | 0 | 0 |
| | Avg. acc. | 41.25 | 66.25 | / | 45.00 | 50.00 |
| LLaVA-v1.6 | Acc. with adv. pairs | 5 | 32.5 | 37.5 | 0 | 0 |
| | Avg. acc. | 51.24 | 56.25 | / | 31.35 | 43.01 |
| MiniGPT-v2 | Acc. with adv. pairs | 0 | 35 | 50 | 10 | 8.82 |
| | Avg. acc. | 47.50 | 62.50 | / | 47.91 | 49.50 |
| LLaVA-Med-v1 | Acc. with adv. pairs | 5 | 45 | 12.5 | 12.5 | 5.88 |
| | Avg. acc. | 43.75 | 68.75 | / | 49.06 | 46.32 |
| LLaVA-Med-v1.5 | Acc. with adv. pairs | 50.00 | 35.00 | 50.00 | 12.5 | 11.76 |
| | Avg. acc. | 72.5 | 62.5 | / | 53.75 | 53.92 |
| BiomedGPT | Acc. with adv. pairs | 0.00 | 5.00 | 50.00 | 10.00 | 2.94 |
| | Avg. acc. | 40.00 | 51.24 | / | 51.04 | 49.01 |
| Med-Flamingo | Acc. with adv. pairs | 2.50 | 45.00 | 50 | 10.00 | 8.82 |
| | Avg. acc. | 43.75 | 72.50 | / | 48.75 | 47.79 |
| CheXagent | Acc. with adv. pairs | 0 | **75** | 50 | 15 | 0 |
| | Avg. acc. | 17.50 | 87.50 | / | 44.58 | 47.05 |
| GPT-4o | Acc. with adv. pairs | **90.00** | 35.89 | **62.50** | 17.94 | **24.24** |
| | Avg. acc. | 93.75 | 65.38 | / | 54.80 | 61.36 |
| GPT-4V | Acc. with adv. pairs | 76.92 | 51.28 | 37.5 | **25.64** | 18.18 |
| | Avg. acc. | 86.25 | 71.79 | / | 58.11 | 61.74 |
| Gemini Pro | Acc. with adv. pairs | 87.5 | **62.5** | 37.5 | 17.5 | 11.76 |
| | Avg. acc. | 91.25 | 77.50 | / | 54.89 | 56.86 |
| | num | 40 | 40 | 8 | 40 | 40 |

## A.7 SPINE MRI

Table 12: Results of different models on Spine MRI in ProbMed. The best-performing model in each question category is **in-bold**, and the second best is underlined.

| | | General Question | | Specialized Question | | |
| --- | --- | --- | --- | --- | --- | --- |
| | | Modality | Organ | Abnormality | Condition/Finding | Position |
| Random Choice | Acc. with adv. pairs | 25 | 25 | 50 | 31.51 | 31.52 |
| LLaVA-v1 | Acc. with adv. pairs | 0 | 32.09 | 50 | 50 | 0.3 |
| | Avg. acc. | 49.22 | 65.58 | / | 47.15 | 49.88 |
| LLaVA-v1.6 | Acc. with adv. pairs | 3.08 | 86.72 | 27.77 | 0.30 | 0 |
| | Avg. acc. | 51.54 | 92.74 | / | 30.59 | 34.37 |
| MiniGPT-v2 | Acc. with adv. pairs | 0.3 | 49.69 | 52.77 | 6.79 | 2.02 |
| | Avg. acc. | 50.15 | 70.52 | / | 48.97 | 50.01 |
| LLaVA-Med-v1 | Acc. with adv. pairs | 1.54 | 5.24 | 36.11 | 12.96 | 5.4 |
| | Avg. acc. | 45.06 | 24.07 | / | 48.52 | 48.04 |
| LLaVA-Med-v1.5 | Acc. with adv. pairs | 70.67 | 84.56 | 50.00 | 11.11 | 11.14 |
| | Avg. acc. | 84.72 | 91.97 | / | 52.40 | 51.89 |
| BiomedGPT | Acc. with adv. pairs | 0.30 | 5.86 | 50.00 | 7.71 | 3.04 |
| | Avg. acc. | 45.06 | 52.77 | / | 51.31 | 44.04 |
| Med-Flamingo | Acc. with adv. pairs | 0.30 | 29.93 | 50 " | 17.90 | 5.40 |
| | Avg. acc. | 50.00 | 64.50 | / | 50.54 | 50.14 |
| CheXagent | Acc. with adv. pairs | 0 | 13.58 | 47.22 | 15.43 | 2.7 |
| | Avg. acc. | 22.53 | 44.44 | / | 51.28 | 48.54 |
| GPT-4o | Acc. with adv. pairs | **98.44** | 84.52 | **63.88** | 19.50 | **24.40** |
| | Avg. acc. | 98.91 | 91.95 | / | 55.46 | 63.70 |
| GPT-4V | Acc. with adv. pairs | 96.28 | **90.71** | 55.55 | 22.6 | 15.59 |
| | Avg. acc. | 97.51 | 94.73 | / | 58.89 | 57.52 |
| Gemini Pro | Acc. with adv. pairs | 98.13 | 88.81 | 57.14 | **24.53** | 14.91 |
| | Avg. acc. | 98.75 | 94.09 | / | 59.19 | 58.20 |
| | num | 332 | 332 | 35 | 332 | 332 |

## A.8 ABDOMINAL MRI

Table 13: Results of different models on Abdominal MRI in ProbMed. The best-performing model in each question category is **in-bold**, and the second best is underlined.

| | | General Question | | Specialized Question | | |
| --- | --- | --- | --- | --- | --- | --- |
| | | Modality | Organ | Abnormality | Condition/Finding | Position |
| Random Choice | Acc. with adv. pairs | 25 | 25 | 50 | 37.13 | 38.26 |
| LLaVA-v1 | Acc. with adv. pairs | 0 | 39.28 | 50.00 | 2.38 | 0 |
| | Avg. acc. | 48.22 | 69.64 | / | 46.42 | 50.00 |
| LLaVA-v1.6 | Acc. with adv. pairs | 2.38 | 73.8 | 35.71 | 1.19 | 0 |
| | Avg. acc. | 51.19 | 85.11 | / | 35.46 | 44.06 |
| MiniGPT-v2 | Acc. with adv. pairs | 0 | 36.9 | 50 | 8.33 | 4.54 |
| | Avg. acc. | 50.00 | 67.26 | / | 47.51 | 51.70 |
| LLaVA-Med-v1 | Acc. with adv. pairs | 2.38 | 47.61 | 50.00 | 14.28 | 9.09 |
| | Avg. acc. | 41.66 | 72.61 | / | 47.42 | 46.46 |
| LLaVA-Med-v1.5 | Acc. with adv. pairs | 51.19 | 65.47 | 50.00 | 13.09 | 16.66 |
| | Avg. acc. | 75.59 | 81.54 | / | 54.31 | 56.37 |
| BiomedGPT | Acc. with adv. pairs | 1.19 | 3.57 | 50.00 | 14.28 | 1.51 |
| | Avg. acc. | 38.69 | 50.00 | / | 51.33 | 46.46 |
| Med-Flamingo | Acc. with adv. pairs | 2.38 | 27.38 | 50.00 | 20.23 | 3.03 |
| | Avg. acc. | 50.59 | 62.50 | / | 49.55 | 50.50 |
| CheXagent | Acc. with adv. pairs | 0 | 26.19 | 50.00 | 11.9 | 10.6 |
| | Avg. acc. | 19.04 | 56.54 | / | 49.20 | 49.62 |
| GPT-4o | Acc. with adv. pairs | **91.66** | 67.85 | 64.28 | 21.42 | **39.39** |
| | Avg. acc. | 95.83 | 81.54 | / | 55.30 | 70.51 |
| GPT-4V | Acc. with adv. pairs | 86.9 | **75** | 50 | 27.38 | 25.75 |
| | Avg. acc. | 92.26 | 85.71 | / | 58.58 | 58.77 |
| Gemini Pro | Acc. with adv. pairs | 89.28 | 72.61 | **85.71** | **28.57** | 25.75 |
| | Avg. acc. | 94.04 | 86.30 | / | 63.39 | 60.98 |
| | num | 84 | 84 | 14 | 84 | 84 |

## A.9 BRAIN X-RAY

Table 14: Results of different models on Brain X-ray in ProbMed. The best-performing model in each question category is **in-bold**, and the second best is underlined.

| | | General Question | | Specialized Question | | |
| --- | --- | --- | --- | --- | --- | --- |
| | | Modality | Organ | Abnormality | Condition/Finding | Position |
| Random Choice | Acc. with adv. pairs | 25 | 25 | 50 | 44.77 | 47.08 |
| LLaVA-v1 | Acc. with adv. pairs | 45.56 | 26.58 | 50 | 0 | 0 |
| | Avg. acc. | 72.78 | 51.89 | / | 48.10 | 50.00 |
| LLaVA-v1.6 | Acc. with adv. pairs | 11.39 | 13.92 | 16.66 | 8.86 | 4.44 |
| | Avg. acc. | 55.06 | 48.10 | / | 45.04 | 48.88 |
| MiniGPT-v2 | Acc. with adv. pairs | 18.98 | **83.54** | 50 | 18.98 | 17.77 |
| | Avg. acc. | 59.49 | 89.87 | / | 51.37 | 52.22 |
| LLaVA-Med-v1 | Acc. with adv. pairs | 8.86 | 8.86 | 0 | 20.25 | 4.44 |
| | Avg. acc. | 54.43 | 31.01 | / | 51.16 | 48.33 |
| LLaVA-Med-v1.5 | Acc. with adv. pairs | 49.36 | 31.64 | 50.00 | 8.86 | 13.33 |
| | Avg. acc. | 73.41 | 56.96 | / | 53.16 | 55.55 |
| BiomedGPT | Acc. with adv. pairs | 12.65 | 6.32 | 50.00 | 11.39 | 2.22 |
| | Avg. acc. | 53.16 | 49.36 | / | 52.95 | 43.33 |
| Med-Flamingo | Acc. with adv. pairs | 8.86 | 0 | 50 | 22.78 | 8.88 |
| | Avg. acc. | 54.43 | 15.18 | / | 50.73 | 48.33 |
| CheXagent | Acc. with adv. pairs | 84.81 | 0 | 50 | 12.65 | 8.88 |
| | Avg. acc. | 92.40 | 29.74 | / | 51.16 | 55.00 |
| GPT-4o | Acc. with adv. pairs | **94.93** | 52.56 | **66.66** | 37.17 | **40.90** |
| | Avg. acc. | 96.20 | 73.71 | / | 62.07 | 69.31 |
| GPT-4V | Acc. with adv. pairs | 82.05 | 8.97 | 33.33 | **43.58** | 22.72 |
| | Avg. acc. | 90.38 | 47.43 | / | 68.48 | 59.09 |
| Gemini Pro | Acc. with adv. pairs | 89.87 | 51.89 | 50 | 31.64 | 31.11 |
| | Avg. acc. | 93.03 | 74.05 | / | 61.81 | 63.88 |
| | num | 79 | 79 | 6 | 79 | 79 |

## A.10 CHEST X-RAY

Table 15: Results of different models on Chest X-ray in ProbMed. The best-performing model in each question category is **in-bold**, and the second best is underlined.

| | | General Question | | Specialized Question | | |
|---|---|---|---|---|---|---|
| | | Modality | Organ | Abnormality | Condition/Finding | Position |
| Random Choice | Acc. with adv. pairs | 25 | 25 | 50 | 37.59 | 37.08 |
| LLaVA-v1 | Acc. with adv. pairs | 28.75 | 36.57 | 50 | 0.12 | 0.11 |
| | Avg. acc. | 64.37 | 68.25 | / | 34.41 | 50.05 |
| LLaVA-v1.6 | Acc. with adv. pairs | 7.11 | 83.97 | 47.94 | 5.89 | 1.52 |
| | Avg. acc. | 53.49 | 91.61 | / | 34.52 | 48.85 |
| MiniGPT-v2 | Acc. with adv. pairs | 4.93 | **94.07** | 50.05 | 18.78 | 11.94 |
| | Avg. acc. | 52.46 | 96.98 | / | 46.09 | 53.15 |
| LLaVA-Med-v1 | Acc. with adv. pairs | 6.25 | 39.77 | 40.24 | 26.28 | 6.14 |
| | Avg. acc. | 52.62 | 67.19 | / | 50.78 | 51.34 |
| LLaVA-Med-v1.5 | Acc. with adv. pairs | 55.44 | 65.48 | 49.53 | 31.82 | 9.78 |
| | Avg. acc. | 77.67 | 82.69 | / | 62.70 | 54.22 |
| BiomedGPT | Acc. with adv. pairs | 91.34 | 86.05 | 50.00 | 16.92 | 9.08 |
| | Avg. acc. | 95.46 | 92.93 | / | 43.00 | 41.46 |
| Med-Flamingo | Acc. with adv. pairs | 80.92 | 90.00 | 50 | 35.83 | 5.24 |
| | Avg. acc. | 90.46 | 95.00 | / | 63.47 | 48.00 |
| CheXagent | Acc. with adv. pairs | 53.68 | 39.64 | **76.59** | **42.75** | 9.38 |
| | Avg. acc. | 76.84 | 69.82 | / | 70.80 | 54.00 |
| GPT-4o | Acc. with adv. pairs | 97.97 | 62.98 | 62.01 | 32.13 | **21.81** |
| | Avg. acc. | 98.81 | 81.39 | / | 59.35 | 59.95 |
| GPT-4V | Acc. with adv. pairs | 91.53 | 67.51 | 53.18 | 39.35 | 21.35 |
| | Avg. acc. | 95.62 | 83.37 | / | 64.69 | 55.64 |
| Gemini Pro | Acc. with adv. pairs | **98.07** | 76.74 | 61.29 | 25.83 | 15.31 |
| | Avg. acc. | 98.94 | 88.32 | / | 52.22 | 54.97 |
| | num | 3120 | 3120 | 1948 | 3120 | 3120 |

## A.11 SPINE X-RAY

Table 16: Results of different models on Spine X-ray in ProbMed. The best-performing model in each question category is **in-bold**, and the second best is underlined.

| | | General Question | | Specialized Question | | |
| | | Modality | Organ | Abnormality | Condition/Finding | Position |
|---|---|---|---|---|---|---|
| Random Choice | Acc. with adv. pairs | 25 | 25 | 50 | 30.95 | 30.99 |
| LLaVA-v1 | Acc. with adv. pairs | 44.55 | 45.04 | 50 | 0.49 | 0 |
| | Avg. acc. | 72.27 | 71.78 | / | 47.32 | 49.42 |
| LLaVA-v1.6 | Acc. with adv. pairs | 4.45 | **82.67** | 33.33 | 1.48 | 0.57 |
| | Avg. acc. | 52.22 | 90.84 | / | 35.87 | 42.02 |
| MiniGPT-v2 | Acc. with adv. pairs | 2.97 | 52.47 | 58.33 | 16.33 | 4.02 |
| | Avg. acc. | 51.48 | 71.78 | / | 53.84 | 51.07 |
| LLaVA-Med-v1 | Acc. with adv. pairs | 8.41 | 7.92 | 33.33 | 17.82 | 5.74 |
| | Avg. acc. | 52.72 | 28.96 | / | 52.82 | 47.58 |
| LLaVA-Med-v1.5 | Acc. with adv. pairs | 46.53 | 71.78 | 50.00 | 14.85 | 13.32 |
| | Avg. acc. | 73.01 | 85.89 | / | 55.78 | 54.79 |
| BiomedGPT | Acc. with adv. pairs | 40.09 | 16.83 | 58.33 | 12.37 | 2.87 |
| | Avg. acc. | 68.06 | 55.19 | / | 50.27 | 40.77 |
| Med-Flamingo | Acc. with adv. pairs | 14.35 | 25.24 | 50 | 14.85 | 5.17 |
| | Avg. acc. | 57.17 | 62.12 | / | 51.09 | 48.38 |
| CheXagent | Acc. with adv. pairs | 82.17 | 20.29 | 62.5 | 16.83 | 0.57 |
| | Avg. acc. | 91.08 | 50.74 | / | 52.70 | 48.70 |
| GPT-4o | Acc. with adv. pairs | **95.54** | **79.70** | 47.82 | **34.15** | **25.86** |
| | Avg. acc. | 97.02 | 89.60 | / | 68.99 | 66.03 |
| GPT-4V | Acc. with adv. pairs | 85.57 | 72.13 | 47.82 | 29.85 | 18.49 |
| | Avg. acc. | 92.03 | 85.32 | / | 65.20 | 57.18 |
| Gemini Pro | Acc. with adv. pairs | 95.02 | 70.14 | **70.83** | 17.91 | 19.07 |
| | Avg. acc. | 96.76 | 84.82 | / | 58.04 | 61.72 |
| | num | 201 | 201 | 24 | 201 | 201 |

## A.12 ABDOMINAL X-RAY

Table 17: Results of different models on Abdominal X-ray in ProbMed. The best-performing model in each question category is **in-bold**, and the second best is underlined.

| | | General Question | | Specialized Question | | |
| --- | --- | --- | --- | --- | --- | --- |
| | | Modality | Organ | Abnormality | Condition/Finding | Position |
| Random Choice | Acc. with adv. pairs | 25 | 25 | 50 | 36.55 | 37.46 |
| LLaVA-v1 | Acc. with adv. pairs | 53.87 | 53.01 | 50 | 2.15 | 0.56 |
| | Avg. acc. | 76.93 | 76.50 | / | 49.14 | 50.00 |
| LLaVA-v1.6 | Acc. with adv. pairs | 5.17 | 56.46 | 46 | 6.46 | 1.12 |
| | Avg. acc. | 52.15 | 75.64 | / | 47.63 | 48.16 |
| MiniGPT-v2 | Acc. with adv. pairs | 4.74 | 38.79 | 50 | 18.53 | 5.64 |
| | Avg. acc. | 52.37 | 67.24 | / | 53.65 | 50.23 |
| LLaVA-Med-v1 | Acc. with adv. pairs | 7.75 | 42.24 | 60 | 14.65 | 4.51 |
| | Avg. acc. | 53.23 | 68.96 | / | 47.47 | 50.87 |
| LLaVA-Med-v1.5 | Acc. with adv. pairs | 52.58 | 50.86 | 50.00 | 6.46 | 14.68 |
| | Avg. acc. | 76.07 | 73.49 | / | 52.02 | 54.75 |
| BiomedGPT | Acc. with adv. pairs | 35.77 | 1.29 | 50.00 | 10.34 | 4.51 |
| | Avg. acc. | 65.30 | 37.50 | / | 52.94 | 46.79 |
| Med-Flamingo | Acc. with adv. pairs | 28.01 | 34.48 | 50 | 14.65 | 4.51 |
| | Avg. acc. | 64.00 | 66.37 | / | 52.52 | 46.25 |
| CheXagent | Acc. with adv. pairs | 77.15 | 23.70 | 70 | 12.93 | 2.25 |
| | Avg. acc. | 88.57 | 52.80 | / | 51.30 | 49.64 |
| GPT-4o | Acc. with adv. pairs | **98.26** | 61.47 | 70 | 27.27 | 21.46 |
| | Avg. acc. | 99.13 | 79.22 | / | 61.83 | 59.81 |
| GPT-4V | Acc. with adv. pairs | 84.84 | 50.21 | 60 | **31.16** | **23.16** |
| | Avg. acc. | 92.42 | 71.42 | / | 59.63 | 57.03 |
| Gemini Pro | Acc. with adv. pairs | 97.14 | **63.36** | **85** | 27.15 | 19.2 |
| | Avg. acc. | 98.70 | 80.81 | / | 59.97 | 58.80 |
| | num | 232 | 232 | 20 | 232 | 232 |

## B DATASET STATISTICS

Table 18: Number of questions across each question type for each image. Ground-truth questions were created based on available metadata, with "yes" answers. For each ground-truth question, we also created a corresponding adversarial question by selecting random adversarial entities and assigning "no" answers. For an image showing a normal organ without abnormality, since there is no ground-truth information on the existence of the condition and position, we only construct hallucinated questions for the condition/finding question type. For an image showing abnormality, the number of question pairs per category equals the number of existing conditions or positions.

| Question type | Image with Normal Organ | Image with Abnormality |
| --- | --- | --- |
| Modality | 2 | 2 |
| Organ | 2 | 2 |
| Abnormality | 1 | 1 |
| Condition/Finding | 1 | 2 x number of existing conditions |
| Position | 0 | 2 x number of existing positions |

## C  IMPACT OF CHAIN-OF-THOUGHT PROMPTS AND VISUAL DESCRIPTIONS ON MODEL PERFORMANCE

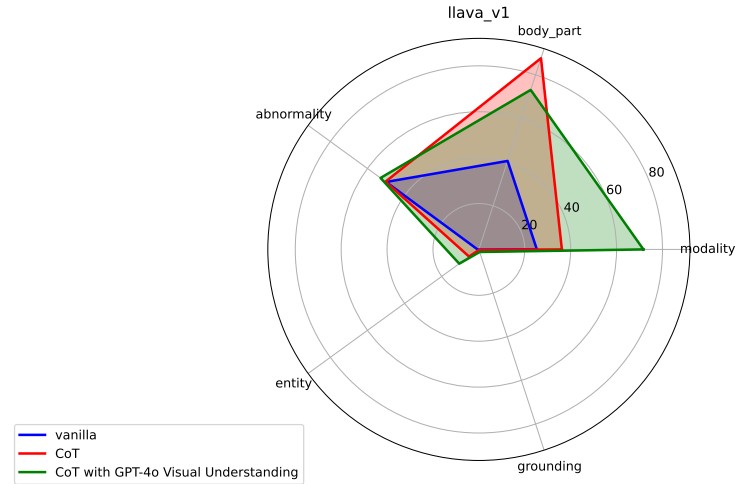

Figure 6: Accuracy of the LLaVA-v1 model across five diagnostic categories under three settings: vanilla (blue), chain-of-thought (CoT, red), and CoT with GPT-4o Visual Understanding (green).

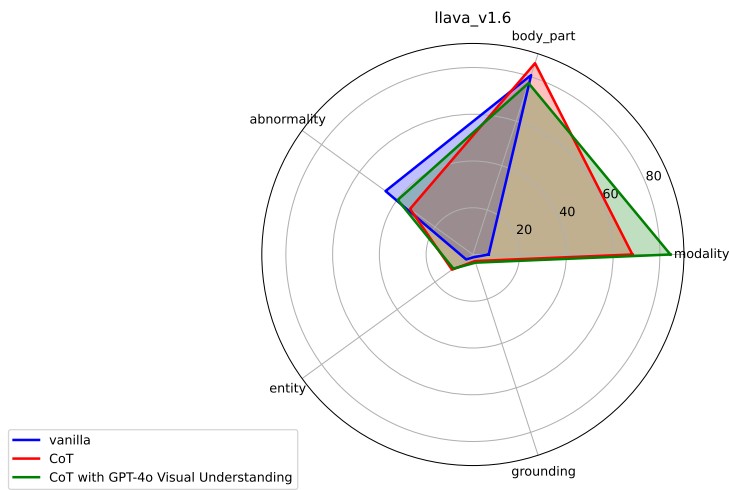

Figure 7: Accuracy of the LLaVA-v1.6 model across five diagnostic categories under three settings: vanilla (blue), chain-of-thought (CoT, red), and CoT with GPT-4o Visual Understanding (green).

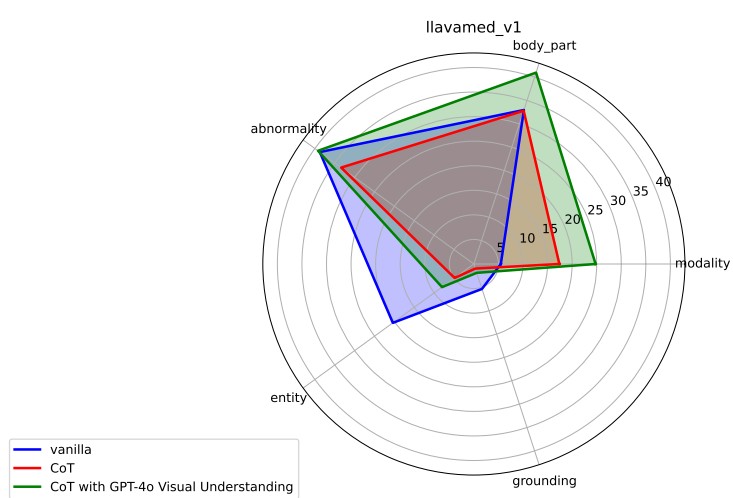

Figure 8: Accuracy of the LLaVA-Med-v1 model across five diagnostic categories under three settings: vanilla (blue), chain-of-thought (CoT, red), and CoT with GPT-4o Visual Understanding (green).

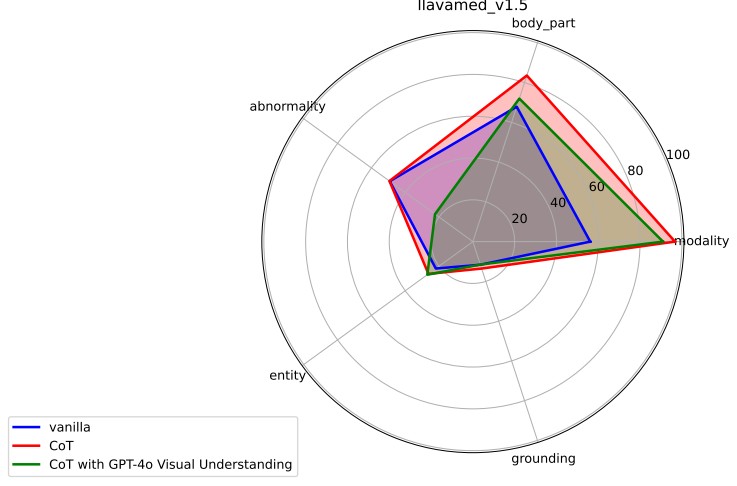

Figure 9: Accuracy of the LLaVA-Med-v1.5 model across five diagnostic categories under three settings: vanilla (blue), chain-of-thought (CoT, red), and CoT with GPT-4o Visual Understanding (green).

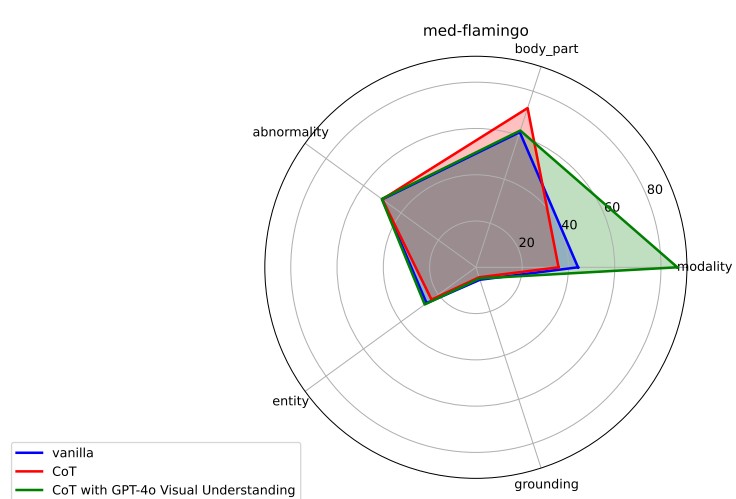

Figure 10: Accuracy of the Med-Flamingo model across five diagnostic categories under three settings: vanilla (blue), chain-of-thought (CoT, red), and CoT with GPT-4o Visual Understanding (green).

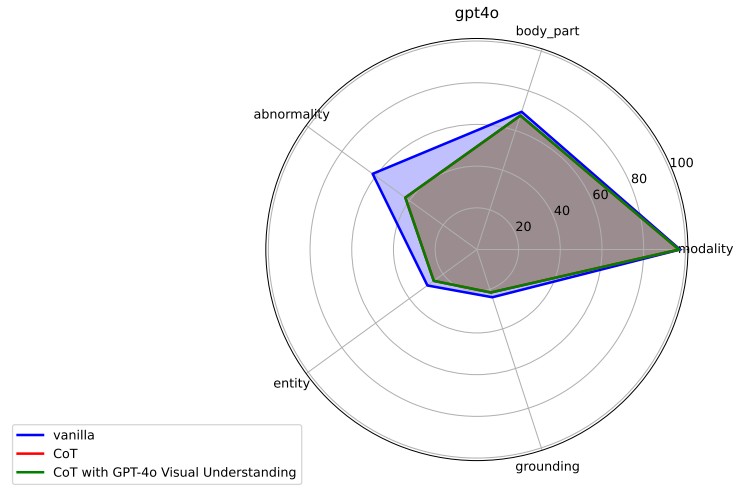

Figure 11: Accuracy of the GPT-4o model across five diagnostic categories under three settings: vanilla (blue), chain-of-thought (CoT, red), and CoT with GPT-4o Visual Understanding (green).

## D   PROMPT DETAILS

The following is the prompt used for extracting medical conditions and their locations from image captions:

```
You are a helpful assistant and you are given a caption describing
    a medical image. Extract medical conditions and diseases,
    along with their locations, if specified. Do not include any
    information that cannot be directly inferred from the image,
    for example, patient status or patient history. Outputs should
     be in the format: "<condition/disease1> : <location1>, <
    condition/disease2> : <location2>...". The term "<location>"
    should include at least one positional descriptor and should
    be explicit in the original caption along with the condition/
    disease. Otherwise, it should be replaced with "None".

For example, consider the caption: "Fig. 1. MRI abdomen and pelvis
     showing the cervical mass." The output should be "<cervical
    mass> : None". For the caption: "Chest radiograph shows
    enlargement of the hilar mass with spread into the left lower
    lobe." The output should be "<enlargement of the hilar mass> :
     <left lower lobe>". Similarly, for the caption: "Abdominal CT
     scan reveals an enhancing rounded pseudo-aneurysm in the
    cystic artery, alongside high-density material within the
    gallbladder's lumen and near the gastrohepatic ligament." The
    correct output is "<enhancing rounded pseudo-aneurysm> : <
    cystic artery>, <high-density material> : <lumen of the
    gallbladder and region of the gastrohepatic ligament>".

Make sure that the response contains only the information in the
    original caption without adding extra details.
```

## E   RESPONSE DISTRIBUTION VISUALIZATION WITHIN EACH CATEGORY

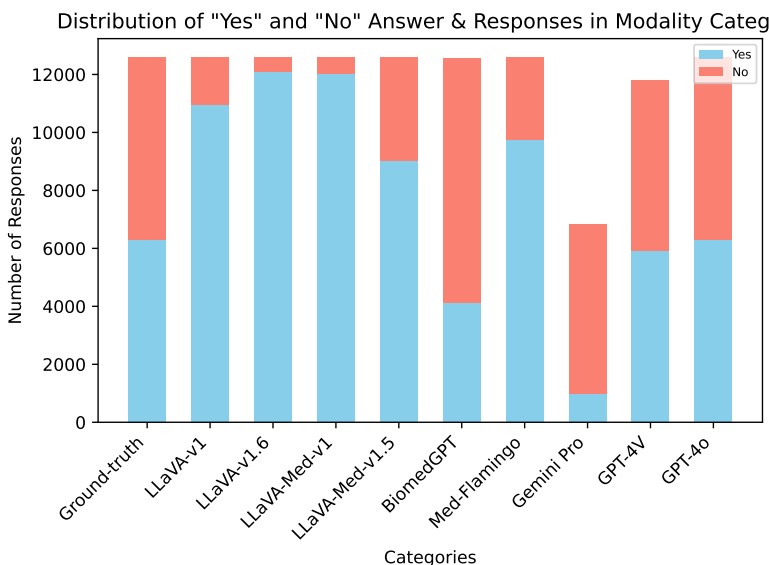

Figure 12: Distribution plot of "yes and "no" ground-truth answers and model responses within the Modality category.

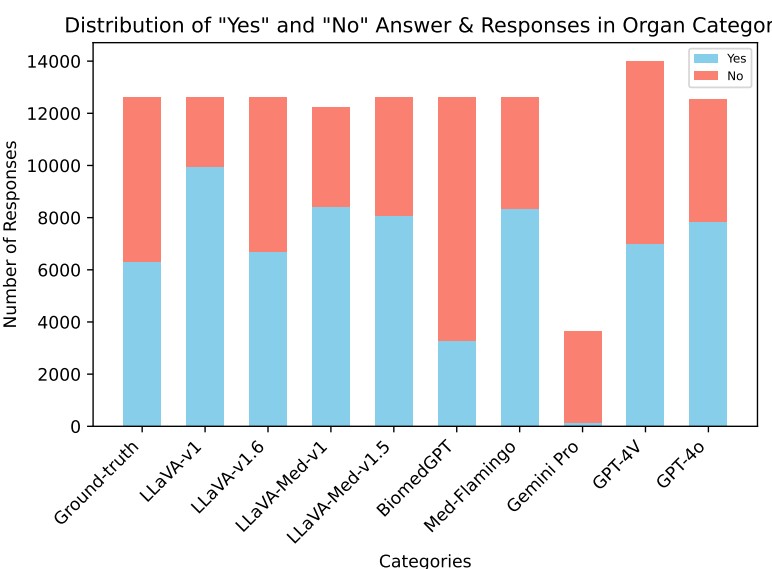

Figure 13: Distribution plot of "yes and "no" ground-truth answers and model responses within the Organ category.

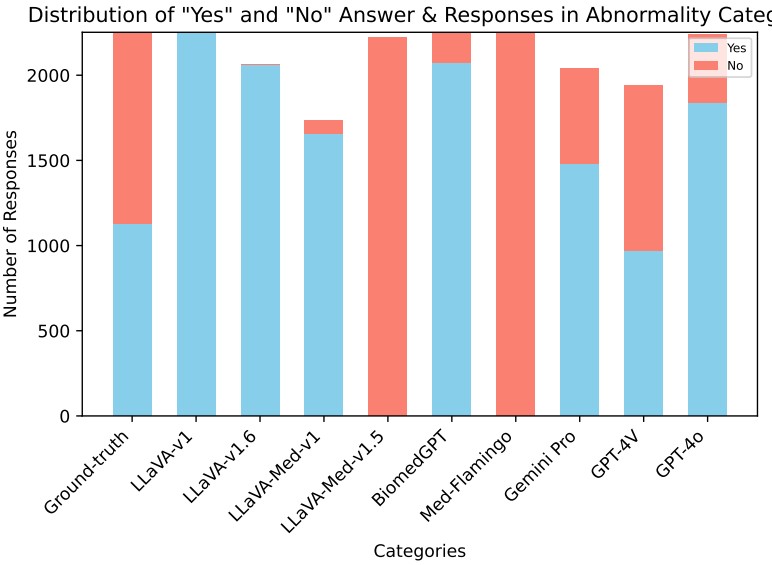

Figure 14: Distribution plot of "yes and "no" ground-truth answers and model responses within the Abnormality category.

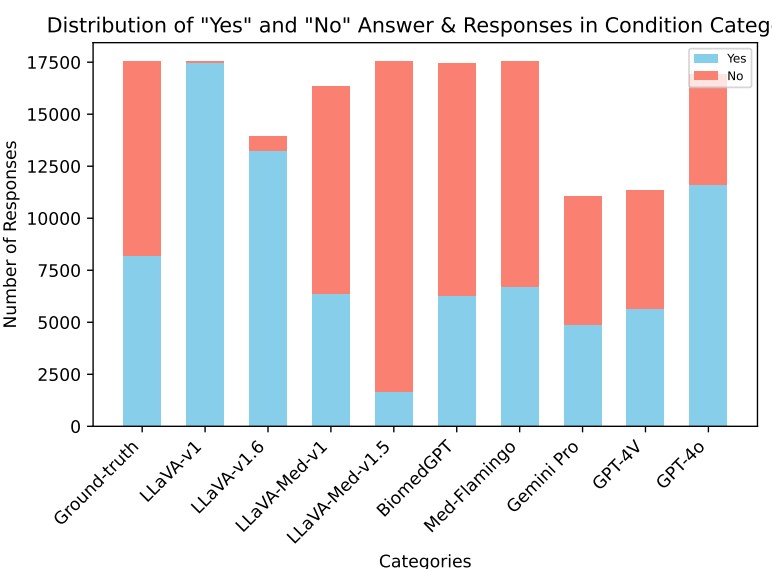

Figure 15: Distribution plot of "yes and "no" ground-truth answers and model responses within the Condition/Finding category.

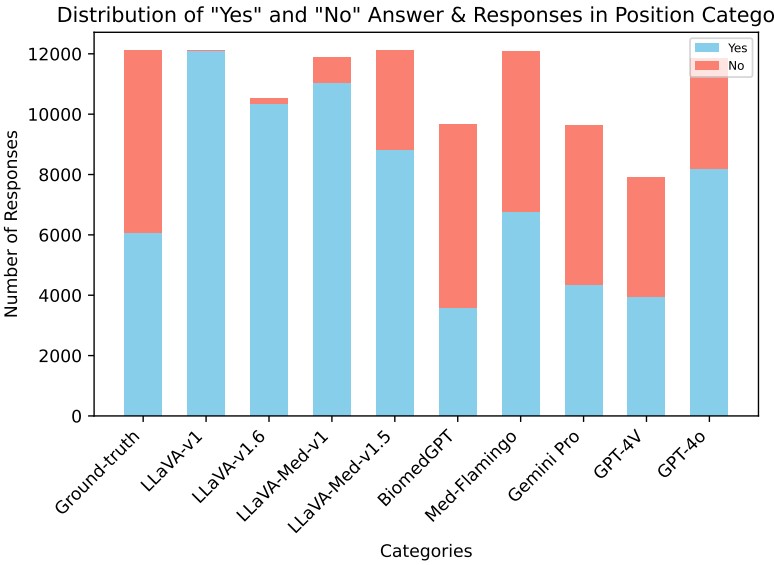

Figure 16: Distribution plot of "yes and "no" ground-truth answers and model responses within the Position category.

