# OpenReview forum: "Worse than Random? An Embarrassingly Simple Probing Evaluation of Large Multimodal Models in Medical VQA"
_ICLR.cc/2025/Conference — Submitted to ICLR 2025_

### Official Review · Reviewer_ivmY · 2024-11-01

**Soundness:** 3
**Presentation:** 3
**Contribution:** 4
**Rating:** 8
**Confidence:** 3

**Summary:**

Authors curated a large, high quality, balanced dataset of 57k close-ended (yes / no) VQA questions for  6,303 radiology medical images sourced from existing open-source datasets. The benchmark can be sliced across a number of different categories and for about half of the questions for which the answer is "yes", an "adversarial" question is also created by perturbing the entities / locations referenced in the original questions and changing the answers to "no". The dataset is used to evaluate a range of closed-source frontier MLLMs as well as open-source general purpose and domain-specific MLLMs.

Overall the authors find that when the adversarial pair is introduced, all models saw significant drops in performance (> 20% in accuracy) although general purpose frontier models like gemini / gpt4o appear to be more robust and saw lower decrease in performance.
The authors further find that when performance is broken down by category, general purpose models can still struggle significantly, and may perform worse on specialized domain-specific MLLMs. Amongst other findings, the authors also investigate whether COT can enhance performance, finding that while open-source models generally saw an improved performance, larger frontier models do not, and open-source models can benefit from using higher quality visual descriptions generated by larger frontier models as part of the COT process, further boosting performance.

**Strengths:**

The dataset is large, high quality, diverse, balanced, and is curated in a methodical manner, with plenty of useful metadata (such as question categorization, location annotation, etc.), making it an incredibly valuable benchmarking resource for the community. With these metadata intact, future works can easily build on top of the released data to curate more open-ended / multiple choice style questions if needed. This in my opinion is the paper's biggest strength. Additionally, the authors share some interesting findings about how much performance degrade when simple adversarial perturbations were applied to the original questions, which shows that even the frontier models today are quite fragile and limited in their ability to perform medical decision making reliably. The investigation into COT, especially COT with external visual descriptions are also quite insightful, revealing a clear path forward to improve open-source, domain-specific MLLMs.

**Weaknesses:**

My main complaint with the paper is that I find the presentation of the results difficult to follow at times. For example, it is not perfectly clear to me whether the "adversarial" questions are included in the tally of 57k QA pairs, and are also included as part of the evaluation benchmark by default - or if the 57k QA pairs represent the default, "clean" benchmark, with additional X number of adversarial questions (presumably = the number of questions with "yes" as answer) used to report the numbers in table 3 and the appendix only. It is also not super clear to me what the authors mean in table 3, where the authors distinguish between "Averaged Accuracy" and "Accuracy (%) with Adversarial Pairs". An illustrative example of how these metrics are computed would be helpful.

Additionally, a limitation of the study is that the benchmark / adversarial questions are only limited to binary yes / no settings, when I can imagine the methodology easily extending to multiple choice questions, which seems like a missed opportunity. Although I believe this is something that future works can build on given the well curated nature of the dataset.

**Questions:**

1. For table 3, my current understanding is that for questions that form a pair of original + adversarial question:
Getting the original question correct, but adversarial question wrong would equate to an accuracy of 50% in the "averaged accuracy", and 0% in the second case. But in either case, the tally of the original questions that do not have an adversarial pair remains consistent? Wouldn't this obfuscate the true impact of the adversarial examples since it's diluted by the original, non-adversarial "no" questions?
2. Are adversarial questions included in the tally of 57k QA pairs, and used in evaluation by default? e.g. in table 4, figure 4, 5, etc.
3. Besides the question + image itself, is there any additional formatting / parsing used to evaluate the various models? especially for open-source MLLMs that may not be perfect at instruction following, what happens if a model outputs a full sentence response instead of a simple yes / no answer.
4. How consistent are these results, especially for larger frontier models where it may not be possible guarantee deterministic output. Is there a way quantify the variance in the performance if e.g. gpt4o / gemini are evaluated more than once?

---

> ### Author Response · Authors · 2024-11-22
> **Official Response to Reviewer ivmY (1/2)**
>
> Thank you for your constructive feedback and thoughtful suggestions. Below, we address your concerns and questions in detail.
>
> **Clarifications on Adversarial Questions and Evaluation Metrics**
>
> *(Addresses Weakness 1 and Questions 1 & 2: Inclusion of adversarial questions and explanation of evaluation metrics)*
>
> `W1: My main complaint with the paper is that I find the presentation of the results difficult to follow at times. For example, it is not perfectly clear to me whether the "adversarial" questions are included in the tally of 57k QA pairs, and are also included as part of the evaluation benchmark by default - or if the 57k QA pairs represent the default.`
>
> `Q1: But in either case, the tally of the original questions that do not have an adversarial pair remains consistent? Wouldn't this obfuscate the true impact of the adversarial examples since it's diluted by the original, non-adversarial "no" questions?`
>
> `Q2: Are adversarial questions included in the tally of 57k QA pairs, and used in evaluation by default? e.g. in table 4, figure 4, 5, etc.`
>
> Adversarial questions are included by default in the 57k VQA pairs and are incorporated into all accuracy metrics reported in the study, except for the "averaged accuracy" column in Table 3. Our evaluation ensures fairness by requiring models to answer both the ground-truth and adversarial questions correctly. For instance, if a model answers the ground-truth question correctly but the adversarial question incorrectly, its "averaged accuracy" in Table 3 would be inflated to 0.5, but its score in our main evaluation framework would remain 0. This design ensures that adversarial questions robustly test model reliability and resilience, as illustrated in Figure 2.
>
> The reported "accuracy with adversarial pairs" in our study is the default metric, ensuring results are not diluted by non-adversarial questions. Table 3 highlights the significant accuracy drop across various models, emphasizing their vulnerability to adversarial challenges and providing insights into model robustness. We have added additional clarifications in Section 4.1.2.
>
> **Why Multiple-Choice Questions Were Not Used**
>
> `W2: Additionally,  a limitation of the study is that the benchmark / adversarial questions are only limited to binary yes / no settings, when I can imagine the methodology easily extending to multiple choice questions, which seems like a missed opportunity.`
>
> As explained in Section 2.2, recent studies [1, 2] highlight key limitations of multiple-choice questions (MCQs), such as selection bias and permutation sensitivity. Specifically:
>
> * A model might select the correct ground-truth attribute (e.g., A from A, B, C) because it finds A relatively more probable than the alternatives, rather than explicitly denying B and C.
> * This behavior complicates the evaluation of a model's reasoning capabilities.
>
> By using binary (yes/no) questions, we adopt a deterministic and straightforward evaluation strategy that ensures models are rigorously tested on their ability to confirm the presence of ground-truth attributes while rejecting hallucinated adversarial attributes.
>
> **Prompting Techniques for Different Model Types**
>
> `Q3: Besides the question + image itself, is there any additional formatting / parsing used to evaluate the various models? especially for open-source MLLMs that may not be perfect at instruction following, what happens if a model outputs a full sentence response instead of a simple yes / no answer.`
>
> For specialized medical-domain models, we followed the prompting formats specified in their respective papers. For general-purpose models, we appended "(please answer yes or no)" to the end of each question. This minor modification aligns with the instruction-following capabilities of general-purpose models, improving response consistency.
>
> When models produced full-sentence responses instead of simple binary answers, we post-processed these outputs to extract the yes/no component for evaluation.

---

> > ### Author Response · Authors · 2024-11-22
> > **Official Response to Reviewer ivmY (2/2)**
> >
> > **Variance in Model Performance and Reproducibility**
> >
> > `Q4: How consistent are these results, especially for larger frontier models where it may not be possible guarantee deterministic output. Is there a way quantify the variance in the performance if e.g. gpt4o / gemini are evaluated more than once?`
> >
> > To assess the consistency of results, we reran experiments using two additional random seeds and reported the updated results (mean and standard deviation) in Table 4. These updates provide a clearer understanding of the variance in model performance, particularly for larger frontier models like GPT-4o and Gemini, where deterministic output is not guaranteed. Including these results ensures the robustness and reproducibility of our findings.
> >
> > We hope these updates address your concerns and enhance the clarity and contributions of our work. Please let us know if additional revisions are needed.
> >
> > [1] Wang, Xinpeng et al. “"My Answer is C": First-Token Probabilities Do Not Match Text Answers in Instruction-Tuned Language Models.” Annual Meeting of the Association for Computational Linguistics (2024).
> > [2] Zheng, C., Zhou, H., Meng, F., Zhou, J., & Huang, M. (2023). Large Language Models Are Not Robust Multiple Choice Selectors. ArXiv, abs/2309.03882.

---

> > > ### Comment · Reviewer_ivmY · 2024-11-26
> > >
> > > Thank you for the additional experiments and the clarification. This helps address my concerns. Although I understand the motivation of using yes/no binary answers only - in my opinion, this is too limiting of a use case for most real-world use cases. Nevertheless, I think this benchmark that provides valuable insight and the overall study is well done. I stand by my original ratings.

---

> > > > ### Author Response · Authors · 2024-11-27
> > > >
> > > > Thank you for your thoughtful feedback and for recognizing the value of our benchmark and study. We appreciate your insights and the time you’ve taken to review our work.

---

### Official Review · Reviewer_9nkQ · 2024-11-02

**Soundness:** 3
**Presentation:** 3
**Contribution:** 3
**Rating:** 6
**Confidence:** 4

**Summary:**

This work curates a medical VQA dataset with pairing original questions and negation questions  and evaluations of SOTA VLMs show the poor visual understanding abilities. Furthermore, the authors show that this issue could be eliminated by adding external visual descriptions generated by GPT-4o.

**Strengths:**

- A new dataset, PromMed, was curated for medical VQA benchmarking, which contains adversarial question pairs
- Comprehensive experiments on multiple VLMs
- Insightful findings: SOTA VLMs perform worse than random guessing on specialized diagnostic questions

**Weaknesses:**

- Some questions are too trivial in the dataset. The ultimate goal of multimodal models is deployment in real clinical practice when the accuracy is good enough. However, no clinician will ask questions on basic modalities (e.g., CT, MR) or organs because they are too trivial. Arguments such as “CheXagent, trained exclusively on chest X-rays, achieved the highest accuracy in determining abnormalities and conditions. However, its performance in general tasks like identifying image modality and organs was lower” should be modified because ChexXagent was not designed for tasks like identifying image modality and organs. These tasks are also not clinically significant because of their trivialities. I would suggest that the authors focus their analysis and discussion on the more clinically relevant and challenging aspects of the dataset, such as identifying specific abnormalities or conditions.

- Minor: References format is not consistent. Some preprint papers even missed arxiv id

**Questions:**

- It is interesting that augmentation with visual descriptions generated by GPT-4o can improve the performance. Can open-sourced multi-modal models (e.g., Qwen2-VL-72b-Instruct, LLaMa 3.2 Vision) improve the performance as well?

- Sec. 4.3.2 it is not clear how CheXagent is used to enhance the model performance. Could you please provide more details on how CheXagent was integrated with other vision-language models?

---

> ### Author Response · Authors · 2024-11-22
> **Official Response to Reviewer 9nkQ (1/2)**
>
> Thank you for your constructive feedback and thoughtful suggestions. Below, we address your concerns and questions in detail.
>
> **Clarifying Dataset Focus and Analysis**
>
> *(Addresses W1: Concern about trivial questions and focus on specialized questions)*
>
> `W1: Some questions are too trivial in the dataset. The ultimate goal of multimodal models is deployment in real clinical practice when the accuracy is good enough. However, no clinician will ask questions on basic modalities (e.g., CT, MR) or organs because they are too trivial. I would suggest that the authors focus their analysis and discussion on the more clinically relevant and challenging aspects of the dataset, such as identifying specific abnormalities or conditions.`
>
> We appreciate your feedback regarding the clinical significance of the questions in our dataset. Our benchmark provides a comprehensive evaluation with 57,000 VQA pairs derived from over 6,000 medical images. Each image is associated with questions ranging from basic attributes, such as organ and modality identification, to more specialized inquiries involving abnormality detection, specific conditions, and their corresponding locations.
>
> While questions about basic modalities or organs may seem trivial, they serve as an essential foundation for assessing a model’s readiness for clinical deployment. Accurate identification of organs and imaging modalities is a critical prerequisite for more advanced diagnostic tasks. Ensuring that models can handle these basic elements is important for their overall reliability in real-life clinical assistance.
>
> In response to your suggestion, we have updated our analysis in Section 4.2.1 to emphasize specialized questions of higher clinical relevance. Sections 4.2.2 and 4.3.2 include in-depth error analyses and procedural diagnostics, focusing on these specialized questions to provide more clinically meaningful insights as well.
>
> **Incorporating Open-Source Multimodal Models**
>
> *(Addresses Q1: Performance improvement with open-source multimodal models in ablation study)*
>
> `Q1: It is interesting that augmentation with visual descriptions generated by GPT-4o can improve the performance. Can open-sourced multi-modal models (e.g., Qwen2-VL-72b-Instruct, LLaMa 3.2 Vision) improve the performance as well?`
>
> We acknowledge the importance of augmenting performance with open-source multimodal models. Due to computational resource constraints, we have extended our experiments to include LLaMA 3.2 Vision (90B) on a subset of 13,132 VQA pairs from the 57,132 VQA pairs in the dataset. The table below compares the performance of four models on the 13k samples under three settings: (i) vanilla inference, (ii) augmentation with external visual descriptions generated by GPT-4o, (iii) augmentation with external visual descriptions generated by LLaMA 3.2 Vision. Similar to the observed improvements with proprietary models like GPT-4o, augmentation with stronger visual descriptions boosts performance.
>
> | Model             | Vanilla | CoT with GPT-4o       | CoT with LLaMA 3.2 Vision (90B) |
> |--------------------|---------|-----------------------|----------------------------------|
> | LLaVA-v1          | 20.36   | **44.08** (+23.72)       | 36.29 (+15.93)                  |
> | LLaVA-v1.6        | 21.84   | **44.52** (+22.68)       | 39.74 (+17.90)                  |
> | LLaVA-Med-v1      | 15.72   | **18.38** (+2.66)        | 14.47 (-1.25)                   |
> | LLaVA-Med-v1.5    | 37.10   | **48.89** (+11.79)       | 41.71 (+4.61)                   |
>
> These findings suggest that the enhancement is not exclusive to proprietary models and that open-source models can also benefit from this approach. Once we complete the evaluation of the full dataset, we will update the results in the paper.

---

> > ### Author Response · Authors · 2024-11-22
> > **Official Response to Reviewer 9nkQ (2/2)**
> >
> > **Clarifications on Section 4.3.2: CheXagent Study**
> >
> > *(Addresses Q2: Clarification on CheXagent’s role in enhancing model performance)*
> >
> > `Q2: Sec. 4.3.2 it is not clear how CheXagent is used to enhance the model performance. Could you please provide more details on how CheXagent was integrated with other vision-language models?`
> >
> > Thank you for pointing out the need for clarification. In Section 4.3.2, we examined whether a model’s expertise in a specific [organ, modality] combination could generalize to unseen modalities. Specifically, we studied CheXagent, a model trained exclusively on chest X-rays, **without integrating it with other vision-language models.**
> >
> > Our findings show that CheXagent achieved higher accuracy in identifying conditions and findings in CT scans and MRIs of the chest - modalities it was not explicitly trained on - compared to other organs within the same unseen modality. The results indicate a 3% increase in accuracy for MRIs and a 4% increase for CT scans. This suggests that CheXagent’s specialized knowledge of the chest can transfer effectively to different imaging modalities of the same organ, highlighting the potential for cross-modal generalization.
> >
> > **Reference Updates**
> >
> > *(Addresses Minor Concern: References format inconsistencies)*
> >
> > We have updated the reference section. Thank you for bringing this to our attention.
> >
> > We hope these updates address your concerns and enhance the clarity and contributions of our work. Please let us know if additional revisions are needed.

---

> > > ### Comment · Reviewer_9nkQ · 2024-11-23
> > >
> > > Thank the authors for the detailed response.
> > >
> > > I appreciate the new experiments on open-source models and clarifications. Most of my concerns have been addressed excepted the data quality since assessing a model’s readiness for clinical deployment needs to use more questions of the clinicians' interest, such as disease diagnosis. However, many of the designed questions are simple modality and organ recognition. I'm pretty sure that no medical professional will ask such obvious questions in clinical practice.
> > > Nevertheless, I'll raise my score properly.

---

> > > > ### Author Response · Authors · 2024-11-23
> > > > **Follow-Up Response to Reviewer 9nkQ**
> > > >
> > > > Thank you for your additional feedback and for raising your score. We greatly appreciate your engagement with our work and would like to further clarify our focus on specialized questions and their importance in this study.
> > > >
> > > > **1. Emphasis on Specialized Questions**
> > > >
> > > > While our dataset includes questions about basic attributes such as organ and modality identification to complete the evaluation pipeline, the focus of our study is on specialized questions that are more clinically relevant. Specifically:
> > > > * Our dataset contains **32k specialized questions** covering abnormality detection, specific conditions, and their corresponding locations. This abundance ensures comprehensive evaluation of clinically significant tasks.
> > > > * The "worse than random" phenomenon referenced in our paper title specifically applies to the specialized questions, highlighting the limitations of current models in these critical areas.
> > > >
> > > > We emphasize specialized questions throughout our analysis. Sections 4.2.2 and 4.3.2 present detailed error analyses and procedural diagnostics focused on these tasks, providing deeper insights into model performance where it matters most.
> > > >
> > > > **2. Broader Importance Beyond Clinical Practice**
> > > >
> > > > Beyond clinical practice and professional use, reliable medical evaluation is critical for people’s daily interactions with large multimodal models in health-related contexts. There is a concerning trend that people start to consult LMMs for medical opinions, potentially due to the “exaggeration” of LMMs’ medical reasoning abilities. In such scenarios, these models must demonstrate robust diagnostic capabilities across specialized tasks to ensure trustworthiness and safety.
> > > >
> > > > * Specialized questions targeting abnormality detection and condition-specific diagnoses are pivotal in ensuring that LMMs provide accurate and meaningful assistance in both professional and layperson use cases.
> > > >
> > > > * Non-clinicians or lay users interacting with LMMs for medical assistance might ask broad, context-establishing questions such as "What type of scan is this?" or "Which organ is shown in this image?" before diving into detailed diagnostic queries. By evaluating models rigorously on these tasks, we aim to contribute to the development of LLMs that can perform reliably in diverse real-world applications.
> > > >
> > > > Ultimately, our benchmark prioritizes specialized questions while providing foundational coverage, enabling holistic evaluation for both professional clinical settings and broader real-world scenarios.

---

### Official Review · Reviewer_vP6d · 2024-11-03

**Soundness:** 3
**Presentation:** 3
**Contribution:** 2
**Rating:** 5
**Confidence:** 4

**Summary:**

The paper introduces the ProbMed dataset for the rigorous evaluation of large multimodal models in clinical domains. The authors show that models perform worse than random on medical diagnosis questions when subjected to adversarial questions. They also show that the poor visual understanding of LMMs is a primary bottleneck, and can be mitigated by adding visual descriptions generated by GPT-4o.

**Strengths:**

It is a well-written paper with many experiments to evaluate the models on the proposed dataset. Several popular models have been studied and also authors propose mitigation strategies to improve the models' performance. Overall, the paper targets important questions and has the potential to be a good contribution to the field.

**Weaknesses:**

After reading the paper, I have a few questions and concerns as follows:


1. (major) If I understood it correctly, the dataset has been curated using the publically available data on the internet. My main concern is the possible data contamination in larger closed or open-source models. When some models have been trained on the data and some not, evaluation using this dataset loses its fairness.

2. If I have gotten it right, all the adversarial questions are in the form of negated questions and their response is "no". Having a model that has been previously trained on the data, can we ensure that it does not cheat?

3. (major) I strongly suggest adding a distribution of the responses that are "yes" or "no". how many questions have an answer of "no" and how many "yes" within each category in the dataset? This can be done through a qualitative distribution plot in the paper.

4. (major) The adversarial question design is creative, but it has issues as well. Within clinical data. there are always cases that have co-occurrence of multiple forms of the disease, but in the original caption, we only have one of them as according to a clinician it is the important one. In this regard, when we create adversarial questions, this important fact has been ignored. So, each question actually needs to be validated by a medical expert. I have seen that 100 samples were examined by experts, but that number is significantly small compared to the size of the dataset. In fact, the paper lacks a thorough and careful expert study to ensure correctness.

**Questions:**

I have mentioned them in weaknesses

---

> ### Author Response · Authors · 2024-11-22
> **Official Response to Reviewer vP6d (1/2)**
>
> Thank you for your thoughtful and constructive feedback. Below, we address your comments and questions in detail.
>
> **Addressing Concerns About Data Contamination and Evaluation Fairness**
>
> *(Addresses W1 and W2: Potential data contamination and model fairness)*
>
> `W1: My main concern is the possible data contamination in larger closed or open-source models. When some models have been trained on the data and some not, evaluation using this dataset loses its fairness.`
>
> `W2: Having a model that has been previously trained on the data, can we ensure that it does not cheat?`
>
> In the era of large multimodal models (LMMs), avoiding data contamination is inherently challenging, especially when datasets are curated from publicly available real-world data. To mitigate this, we deliberately introduced adversarial question-answer pairs alongside ground-truth pairs to enhance robustness and fairness.
>
> For ground-truth QA pairs, we transformed the data into formats that differ from their original representations. For example:
> * **ChestX-ray14:** Bounding boxes were converted into textual positional descriptors, and classification labels were integrated into our VQA curation process.
> * **MedICaT:** Captions were processed into structured metadata containing extracted ground-truth information rather than being directly used.
>
> Each ground-truth QA pair was paired with an adversarial counterpart using hallucinated attributes. These adversarial attributes were carefully curated to be unseen by models, even those potentially trained on the original data. Additionally, our evaluation framework calculates accuracy by requiring models to correctly answer both the ground-truth and adversarial pairs. As illustrated in Figure 2, if a model answers the ground-truth question correctly but the adversarial one incorrectly, it receives no credit. This mechanism ensures robustness and prevents inflated scores due to data contamination.
>
> The drop in accuracy in Table 3 highlights the vulnerability of models to adversarial perturbations. The scores reported elsewhere in the paper are “accuracy with adversarial pairs” by default.
>
> **Ensuring Robustness and Quality of Adversarial QA Pairs**
>
> *(Addresses W4: Concerns about adversarial question quality and expert validation)*
>
> `W4: The adversarial question design is creative, but it has issues as well. Within clinical data. there are always cases that have co-occurrence of multiple forms of the disease, but in the original caption, we only have one of them as according to a clinician it is the important one. In this regard, when we create adversarial questions, this important fact has been ignored. So, each question actually needs to be validated by a medical expert. I have seen that 100 samples were examined by experts, but that number is significantly small compared to the size of the dataset. In fact, the paper lacks a thorough and careful expert study to ensure correctness.`
>
> The metadata underlying our dataset comes from expert-verified sources such as ChestX-ray14 and MedICaT, which inherently account for the co-occurrence of multiple diseases in a single medical image. After preprocessing, 40% of the images marked as abnormal in our dataset feature multiple conditions. This information was derived from:
> * Multiple labels in ChestXray14.
> * The presence of several disease names in MedICaT captions.
>
> To comprehensively evaluate models, we generated two QA pairs for each condition in a single image. Adversarial QA pairs were constructed by selecting random attributes within the same organ-modality category (e.g., conditions for a Chest X-ray were sourced from conditions identified in other Chest X-rays). We ensured that adversarial attributes did not overlap with the existing condition(s) in the image. This careful design guarantees that adversarial QA pairs are logically consistent and robust.
>
> If a condition was omitted in the original data source, the likelihood of selecting it as a hallucinated attribute was minimal - estimated at 0.53% across all [organ, modality] combinations (Table 2).
>
> To further validate the correctness of metadata and QA pairs, we engaged two medical experts - a postdoctoral researcher and a graduate student from a U.S. medical school - to review 100 randomly sampled metadata entries and the corresponding **1,090** curated QA pairs (including adversarial QA pairs). The expert reviewers verified two aspects for each sample: (i) whether the “condition: location” field in the curated metadata is valid given the original caption and (ii) if the question-answer pairs are valid given the original [image, caption]. Their evaluations yielded an average metadata accuracy of 94.00% and QA pair accuracy of 97.79%, underscoring the reliability of our dataset.

---

> > ### Author Response · Authors · 2024-11-22
> > **Official Response to Reviewer vP6d (2/2)**
> >
> > **Adding Response Distribution Visualization**
> >
> > *(Addresses W3: Suggestion to include distribution of “yes” and “no” responses)*
> >
> > `W3: I strongly suggest adding a distribution of the responses that are "yes" or "no". how many questions have an answer of "no" and how many "yes" within each category in the dataset? This can be done through a qualitative distribution plot in the paper.`
> >
> > To address your request, we have included qualitative distribution plots in the revised paper (Appendix E). These visualizations shows:
> > 1. The balanced distribution of "yes" and "no" answers within each category in the dataset.
> > 2. The distribution of model responses across each of the question categories.
> >
> > These plots clarify the dataset's balanced answers and enhance the transparency of our evaluation framework.
> >
> > We hope these clarifications and updates address your concerns and strengthen the contributions of our work. Please let us know if additional revisions or explanations are needed.

---

> ### Author Response · Authors · 2024-11-24
> **Follow-Up Response to Reviewer vP6d (1/2)**
>
> Thank you for your follow-up comments and for engaging with our work. We appreciate the opportunity to address your concerns further and clarify the new plots.
>
> **1. Addressing Data Contamination Concerns**
>
> We acknowledge the difficulty in fully eliminating the possibility of data contamination, particularly for proprietary API-based models whose pretraining data is not disclosed. However, the design of ProbMed mitigates this issue through our rigorous adversarial QA framework.
>
> Specifically:
>
> * Each ground-truth QA pair is paired with an adversarial counterpart constructed from random selection, requiring models to correctly answer both the ground-truth and adversarial questions to earn credit. This approach penalizes reliance on memorized patterns and ensures robustness against potential contamination.
>
> * As shown in **Table 3**, all models experience a significant accuracy drop (on average 29.97%) across all models when adversarial pairs are introduced, indicating that performance is not solely reliant on prior exposure to the data but rather on their reasoning and generalization abilities. If a model were simply memorizing ground-truth infomation, it would not have exhibited the **worse-than-random** performance we observed for **specialized** questions.
>
> It is also important to note that the use of publicly available sources as data, even for test sets, is not unique to ProbMed. Many well-cited benchmarks face similar challenges:
>
> * [1] sources data from publicly available materials, including practice questions for exams like the Graduate Record Examination (GRE) and the United States Medical Licensing Examination (USMLE).
> * [2] uses data drawn from online resources, textbooks, and lecture materials.
> * Another highly regarded medical benchmark, [3], derives its test sets from pathology textbooks and online digital libraries.
>
> In this era, any publicly available source could potentially serve as training material for LMMs, particularly proprietary API-based models. While this reality poses inherent limitations, it reinforces the importance of rigorous evaluation frameworks like ProbMed to ensure fairness, minimize bias, and accurately measure models' reasoning capabilities.
>
> **2. Clarifying the Total Number of Responses in Distribution Plots**
>
> The discrepancy in the total number of responses across different models arises from invalid answers, such as when models refuse to answer or generate irrelevant content. These invalid responses are considered incorrect in our accuracy calculation and excluded from valid responses' distribution plots.
>
> This phenomenon is most common among API-based proprietary models. For example, in Section 4.2.2, we analyze errors for models such as GPT-4V and Gemini Pro. Table 5 in the paper provides a breakdown of errors into categories, including:
>
> * **Deny Ground Truth:** Incorrectly answering "no" to ground-truth questions.
> * **Accept Hallucination:** Incorrectly answering "yes" to adversarial questions.
> * **Reject to Answer:** Cases where the model either refuses to answer or generates irrelevant content.
>
> The distribution plots focus solely on direct valid responses to better illustrate the models' performance characteristics. When models produced full-sentence responses instead of simple binary answers, we post-processed these outputs by letting GPT-4 decide the correctness of the answer. We hope this explanation clarifies the observed differences in the total number of responses across models.
>
> We appreciate your thoughtful feedback and the opportunity to clarify these points. If there are additional questions or further clarifications needed, we are happy to provide them. Thank you again for your engagement and support.
>
> [1] Hendrycks, Dan et al. “Measuring Massive Multitask Language Understanding.” ArXiv abs/2009.03300 (2020): n. pag.
>
> [2] Yue, Xiang et al. “MMMU: A Massive Multi-Discipline Multimodal Understanding and Reasoning Benchmark for Expert AGI.” 2024 IEEE/CVF Conference on Computer Vision and Pattern Recognition (CVPR) (2023): 9556-9567.
>
> [3] He, X., Zhang, Y., Mou, L., Xing, E.P., & Xie, P. (2020). PathVQA: 30000+ Questions for Medical Visual Question Answering. ArXiv, abs/2003.10286.

---

> ### Author Response · Authors · 2024-11-24
> **Follow-Up Response to Reviewer vP6d (2/2)**
>
> **3. Proposed Categorization of Models by Training Data Transparency**
>
> To address your concerns further, we propose a framework to enhance the transparency of model evaluations on ProbMed by categorizing models based on the disclosure and nature of their training data. This approach aims to provide clearer insights into the potential influence of data exposure on model performance and promote fairer comparisons. For example, a potential categorization could be:
>
> **a. Proprietary/Undisclosed Training Data**
>
> Models in this category rely on proprietary datasets with undisclosed training sources. This lack of transparency introduces the highest uncertainty regarding data contamination. Examples include:
> * OpenAI's GPT models
> * Google's Gemini models
>
> **b. Partially Disclosed Training Data**
>
> These models provide partial transparency regarding their training data, listing some sources but not the full scope. Examples include:
> * Meta's LLaMA models trained on hybrid datasets that combine publicly available and proprietary sources
>
> **c. Fully Open Source with Known Data Sources**
>
> This category includes models that are fully open source, with complete disclosure of training datasets and pipelines. Examples include:
>
> * Academic models published with detailed training protocols, such as BiomedGPT
>
> **d. Fine-Tuned Models (Domain-Specific)**
>
> These models are fine-tuned on specialized datasets after their general pretraining phase, often targeting specific domains such as medicine. Examples include:
>
> * LLaVA-Med and Med-Flamingo, fine-tuned LMMs on medical QA datasets
>
> **Implementation on the Leaderboard**
>
> To ensure fairness and transparency, we propose implementing this categorization on the ProbMed leaderboard:
>
> 1. **Label each model with its training data category** alongside its performance metrics, especially for proprietary models that lack of disclosure of training resources.
> 2. **Indicate any fine-tuning on medical datasets** or domain-specific datasets to differentiate performance improvements due to further customization.
>
> **How This Addresses Data Contamination Concerns**
>
> This categorization will not eliminate the possibility of data contamination (which is nearly impossible for all publicly available datasets and benchmarks) but will:
>
> * Provide transparency to users of the benchmark about the degree of training data overlap risk.
> * Facilitate fairer comparisons by contextualizing performance metrics based on data exposure.
> * Encourage future discussions and advancements in benchmarking practices to minimize bias and contamination.
>
> We believe this proposal strikes a balance between acknowledging inherent challenges in evaluating LLMs and ensuring robust and transparent comparisons. If additional details or modifications are needed, we would be happy to refine this approach further.

---

> > ### Author Response · Authors · 2024-11-26
> > **Looking forward to your response**
> >
> > Dear Reviewer vP6d,
> >
> > Thank you for your valuable feedback. We have clarified the robustness of our adversarial QA design in mitigating data contamination concerns, explained the distribution plots and error handling, and proposed a categorization framework for models based on the transparency of their training data. This framework aims to provide clearer insights into the potential influence of data exposure on model performance and promote fairer comparisons.
> >
> > We kindly invite you to revisit our paper in light of these updates. We hope these improvements address your concerns and encourage a reevaluation of your rating.
> >
> > Best,
> >
> > The Authors

---

> ### Author Response · Authors · 2024-12-03
> **Follow-Up Response to Reviewer vP6d**
>
> Thank you for your constructive feedback and for continuing to engage with our work. We value your thoughtful suggestions and your acknowledgment of the dataset's contributions to the field. Below, we provide additional clarifications and new result analysis (as suggested) to address your concerns about dataset bias and contamination risks.
>
> 1. **Empirical Experimentation with Bias Analysis**
>
> In response to your suggestion and within the constraints of time and computational resources, we conducted an additional experiment to empirically examine the potential bias introduced by dataset overlap. Specifically, we fine-tuned **Phi-3.5-Vision**, a lightweight **4.2B** open-source multimodal model with strong instruction-following capability, using data from the MedICaT dataset and conducted the following comparative analysis:
>
> **Training Settings:**
>
> (i) **Clean data:** 20K clean examples from MedICaT, ensuring no overlap with the 4.5K MedICaT images included in ProbMed.
>
> (ii) **Contaminated Data:** 20K examples that include all the 4.5K images used in constructing ProbMed (22.5% overlap).
>
> **Evaluation Data:**
>
> (i) All models were evaluated on the 41.8K VQA pairs in ProbMed curated from MedICaT.
>
> (ii) To show the effectiveness of the adversarial evaluation, we also evaluated the models without adversarial pairs for comparison.
>
> **Findings:**
>
> * The comparison of zero-shot vs. in-domain fine-tuning on unseen MedICaT data reflects a typical and expected outcome. In-domain fine-tuning inherently helps the model align better with the task-specific distributions, as demonstrated by the jump from **43.97%** (zero-shot) to **46.21%** (Fine-tuned on clean, unseen data).
>
> * However, compared with fine-tuning on the **clean, unseen MediCaT data**, fine-tuning directly on the **contaminated data** only brings **1.65%** improvement on ProbMed, which shows the robustness of our adversarial evaluation benchmark ProbMed in evaluating the reliability of LMMs in MedVQA. ProbMed’s rigorous adversarial QA framework is designed to stress-test reasoning and generalization of LMMs (e.g., the LMM has to answer all positive and negative questions about the same image correctly to count as a hit), reducing reliance on memorized patterns. To further verify this, we evaluated the “contaminated” model on normal medical VQA data in ProbMed **without adversarial pairs**, the accuracy is as high as **72.24% (with an 11.40% absolute improvement vs. 1.65% on ProbMed).**
>
> * In summary, based on the quick evaluation of Phi-3.5-Vision model, fine-tuning a LMM within the same domain provides some advantages; while data contamination certainly leads to better performance, its impact is effectively mitigated by the adversarial evaluation mechanism in ProbMed.
>
> | Evaluation \ Training                           | Zero-shot | Finetuned on 20k MediCaT (clean, unseen) | Finetuned on 20k MediCaT (including 4.5k contaminated) |
> |------------------------------------------------|-----------|------------------------------------------|---------------------------------------------------------|
> | ProbMed                                       | 43.97     | 46.21                                    | 47.86 (+1.65)                                           |
> | ProbMed **without Adversarial Pairs**         | 50.21     | 60.84                                    | 72.24 (+11.40)                                          |
>
> *Phi-3.5-vision-instruct fine-tuning parameters: Epoch = 1, lr = 4.0e-5, bs = 16, wd = 0.01, LoRA rank = 64*
>
> 2. **Potential Bias Discussion and Model Categorization**
>
> The categorization of LMMs based on training data sources is also very important based on the results. To further improve transparency, we propose categorizing models based on their transparency in training data. We will include a Discussion section in the revision to delve deeper into these topics, including the influence of dataset contamination and the role of benchmarks like ProbMed in advancing reliable evaluations.
>
> ProbMed’s rigorous adversarial QA framework is designed to stress-test reasoning and generalization, reducing reliance on memorized patterns. However, we acknowledge that adversarial designs cannot fully eliminate data contamination risks, especially for proprietary models with undisclosed training datasets. While ProbMed is not a definitive measure of clinical applicability, it serves as a robust benchmark to evaluate LMM reasoning under adversarial conditions.
>
> We appreciate your recommendation for such an analysis and recognize its value in contextualizing the strengths and limitations of ProbMed. We will aim to extend this analysis further in future work to quantify dataset bias comprehensively.

---

### Official Review · Reviewer_6yHr · 2024-11-03

**Soundness:** 2
**Presentation:** 2
**Contribution:** 2
**Rating:** 3
**Confidence:** 4

**Summary:**

This paper introduces ProbMed, a Med-VQA dataset with adversarial negative samples to reveal multimodal models' limitations in complex medical tasks. Chain-of-thought reasoning and GPT-4-generated visual descriptions are shown to enhance fine-grained diagnostic performance.

**Strengths:**

1. By introducing adversarial negative samples, the dataset tests model robustness, offering a more challenging evaluation standard.
2. ProbMed includes various diagnostic tasks across different imaging modalities and body organs, providing a evaluation setting for models.
3. The model’s performance improvement strategies are validated through methods like chain-of-thought reasoning and visual description enhancement, providing a foundation for future advancements.

**Weaknesses:**

1. The dataset has a significantly higher number of chest X-ray images than other types, which may lead to poorer model performance on other organs. Has the model's performance across different modalities been experimentally verified?
2. How does the study prevent hallucinations when using GPT-4 for caption analysis, abnormality identification, and positional descriptions through few-shot prompting?
3. ProbMed uses closed-ended questions and lacks open-ended tasks, limiting the dataset's comprehensiveness for evaluation.
4. Constructing adversarial samples by selecting random entities (such as alternative organs, modalities, or hallucinated conditions with "no" answers) may increase testing difficulty but could also introduce mistakes, resulting in samples that aren’t truly adversarial.

**Questions:**

1. Is the construction of adversarial negative samples reasonable? Could additional strategies be introduced to ensure the validity of these adversarial samples?
2. Does class imbalance in the dataset affect the model's generalization ability? How can better diagnostic performance be achieved for organs beyond the chest? Additionally, validation across multiple modalities is recommended to assess the impact of imbalanced data distribution on each modality.
3. What is the impact of adding open-ended tasks on the model's performance? How can report generation or open-ended question-answering be incorporated into ProbMed? Clearly, open-ended questions are more representative of common clinical scenarios.
4. Could GPT-4-generated visual descriptions introduce bias or hallucinations?

---

> ### Author Response · Authors · 2024-11-22
> **Official Response to Reviewer 6yHr (1/2)**
>
> Thank you for your thoughtful and constructive feedback. Below, we address your comments and questions in detail.
>
> **Dataset Balance and Generalization Across Modalities**
>
> *(Addresses W1 and Q2: Concerns about dataset balance and verification across modalities)*
>
> `Q2: Does class imbalance in the dataset affect the model's generalization ability?`
>
> Our benchmark provides a comprehensive and rigorous evaluation of 57k VQA pairs derived from over 6k medical images spanning twelve [organ, modality] combinations. While public Chest X-ray data is more abundant, the dataset features balanced binary answers. Similar to widely cited benchmark papers in the field [1,2], we focused on **pure evaluation without model fine-tuning**, adhering to the standard research practice of evaluating pre-trained models. Hence, our approach does not compromise the models' generalization ability.
>
> `W1: Has the model's performance across different modalities been experimentally verified?`
>
> `Q2: Validation across multiple modalities is recommended to assess the impact of imbalanced data distribution on each modality.`
>
> Breakdown results for each of the twelve [organ, modality] combinations are provided in Appendix A. Consistent with these subsets, our analysis in Section 4 demonstrates robust findings across categories. Furthermore, the ablation study (4.3.1 and Appendix C) reveals universal performance improvements across all combinations, further highlighting the robustness of our evaluation design.
>
> **Metadata Extraction and Adversarial QA Construction**
>
> *(Addresses W2, W4, and Q1: Hallucination prevention, adversarial question quality, and expert validation)*
>
> `W2: How does the study prevent hallucinations when using GPT-4 for caption analysis, abnormality identification, and positional descriptions through few-shot prompting?`
>
> The metadata used to construct our dataset originates from expert-verified ChestX-ray14 and MedICaT datasets. GPT-4's role was limited to extracting condition names and their corresponding positions from MedICaT captions through few-shot prompting (Appendix D). This textual extraction task is significantly simpler than diagnosing from medical images and poses minimal risks of hallucination.
>
> `W4: Constructing adversarial samples by selecting random entities (such as alternative organs, modalities, or hallucinated conditions with "no" answers) may increase testing difficulty but could also introduce mistakes, resulting in samples that aren’t truly adversarial.`
>
> `Q1: Is the construction of adversarial negative samples reasonable? Could additional strategies be introduced to ensure the validity of these adversarial samples?`
>
> To construct adversarial QA pairs, we employed a rigorous process:
> 1. **General Questions:** Adversarial attributes (e.g., organs or modalities) were logically invalidated by filtering the original [image, caption] pairs to ensure the image contained a single organ of a single modality.
> 2. **Specialized Questions:** Adversarial attributes were selected within the same [organ, modality] category to enhance challenge and robustness while ensuring no overlap with existing condition(s) in the image.
>
> By design, adversarial QA pairs in our dataset are logically consistent and challenging yet robust. That being said, if a disease was indeed omitted in the original data source, the likelihood of the omitted disease being selected as a hallucinated attribute is minimal - estimated at 0.53% on average across all [organ, modality] combinations, according to statistics in Table 2.
>
> An expert verification process further confirmed metadata and QA pair quality (Section 3.3). Two annotators - a postdoctoral researcher and a graduate student from a U.S. medical school - reviewed 100 randomly sampled metadata and 1,090 corresponding QA pairs (including adversarial QA pairs). The annotators were instructed to verify two aspects for each sample: (i) whether the “condition: location” field in the curated metadata is valid given the original caption and (ii) if the question-answer pairs are valid given the original [image, caption]. Their evaluations yielded an average metadata accuracy of 94.00% and QA pair accuracy of 97.79%, underscoring the reliability of our dataset.

---

> ### Author Response · Authors · 2024-11-22
> **Official Response to Reviewer 6yHr (2/2)**
>
> **Structured Probing vs. Open-Ended Evaluation**
>
> *(Addresses W3 and Q3: Why closed-ended tasks were chosen and their scalability)*
>
> `W3: ProbMed uses closed-ended questions and lacks open-ended tasks, limiting the dataset's comprehensiveness for evaluation.`
>
> `Q3: What is the impact of adding open-ended tasks on the model's performance? How can report generation or open-ended question-answering be incorporated into ProbMed?`
>
> Open-ended tasks, such as medical report generation, pose significant challenges for LMMs due to hallucinations and the complexity of evaluating free-text outputs. These tasks often require labor-intensive, subjective human evaluation, making them difficult to scale [3]. In contrast, our approach emphasizes structured binary (yes/no) questions, which offer a scalable, deterministic framework for fine-grained probing. This structured approach ensures consistency and stability while effectively evaluating LMMs in clinically relevant scenarios. Additionally, our dataset includes inline captions for each image, enabling future research to extend evaluation to open-ended tasks if desired.
>
> **Rationale Behind the Ablation Study**
>
> *(Addresses Q4: Insights from augmenting open-source models with GPT-4 visual reasoning)*
>
> `Q4: Could GPT-4-generated visual descriptions introduce bias or hallucinations?`
>
> The ablation study in Section 4.3 investigates how proprietary models like GPT-4 can improve the performance of open-source, domain-specialized models. While GPT-4's performance remains imperfect, with limitations like hallucinations, it consistently outperformed other models (Table 4, Appendix A). Our findings reveal an average accuracy improvement of 9.44% across all question categories when open-source models are augmented with GPT-4's visual reasoning capabilities. This highlights the critical role of enhancing visual understanding to address bottlenecks in the performance of existing medical LMMs.
>
> We hope this addresses your concerns. Please let us know if further clarification is needed.
>
> [1] Yue, Xiang et al. “MMMU: A Massive Multi-Discipline Multimodal Understanding and Reasoning Benchmark for Expert AGI.” 2024 IEEE/CVF Conference on Computer Vision and Pattern Recognition (CVPR) (2023): 9556-9567.
>
> [2] Hendrycks, Dan et al. “Measuring Massive Multitask Language Understanding.” ArXiv abs/2009.03300 (2020): n. pag.
>
> [3] Chaves, Juan Manuel Zambrano, et al. "Towards a clinically accessible radiology foundation model: open-access and lightweight, with automated evaluation." arXiv preprint arXiv:2403.08002 (2024).

---

> ### Author Response · Authors · 2024-11-26
> **Looking forward to your response**
>
> Dear Reviewer 6yHr,
>
> Thank you for your valuable feedback. We have clarified the dataset balance, adversarial QA construction process and validation, and highlighted the structured evaluation framework's scalability.
>
> We kindly invite you to revisit our paper in light of these updates. We hope these improvements address your concerns and encourage a reevaluation of your rating.
>
> Best,
>
> The Authors

---

> > ### Comment · Reviewer_6yHr · 2024-11-27
> >
> > The authors’ response fails to address the core issue. While they emphasize the significant challenges posed by open-ended tasks for LMMs, challenges should not be an excuse to overlook real clinical scenarios. Regarding the clinical application of Med-VQA, are we expected to provide multiple answer options for every question posed to the VQA system? Is this practical for clinical use? This is precisely why previous datasets like RAD, SLAKE, and PathVQA distinguish between closed-end and open-end questions.

---

> ### Author Response · Authors · 2024-11-27
> **Follow-Up Response to Reviewer 6yHr**
>
> Thank you for your continued engagement. We would like to clarify the scope and contributions of our work since there seems to be some confusion about that. **ProbMed is not intended to replace existing benchmarks for MedVQA but to address a critical gap in reliably evaluating LMMs within MedVQA.** Its unique incorporation of adversarial negation pairs for each question-answer pair ensures diagnostic specificity and reliability, setting ProbMed apart from the existing closed-ended MedVQA framework. For instance, Table 3 highlights significant performance drops when our adversarial pairs are introduced to the VQA-RAD test set, demonstrating the limitations of current models that reported near-saturated results in earlier studies [1, 2]. Additionally, it enables categorical accuracy assessments across various procedural diagnostic dimensions for each image and allows fine-grained analysis.
>
> Our structured approach, using closed-ended questions, ensures quantifiable and consistent evaluations, revealing insights that less controlled settings might miss. While ProbMed does not currently encompass the complexities of open-ended real-world clinical scenarios, it lays a critical foundation for future research that can extend to include such elements. Our goal is to enhance the understanding of LMM capabilities and limitations in medical imaging, complementing - not replacing - other benchmarks and paving the way for more clinically realistic tasks as technologies evolve. We hope this clarifies the scope and contributions of our work.
>
> [1] Li, Chunyuan, et al. "Llava-med: Training a large language-and-vision assistant for biomedicine in one day." Advances in Neural Information Processing Systems 36 (2024).
>
> [2] Moor, Michael, et al. "Med-flamingo: a multimodal medical few-shot learner." Machine Learning for Health (ML4H). PMLR, 2023.

---

### Meta-Review · Area_Chair_a1wo · 2024-12-23

**Metareview:**

This paper introduces ProbMed, a curated Med-VQA dataset with 57k balanced yes/no questions and adversarial negatives to assess multimodal models' diagnostic capabilities, revealing their poor visual understanding and showing that GPT-4-generated visual descriptions and chain-of-thought reasoning can improve performance.


Reviewers commended the ProbMed dataset for its high quality, diversity, and adversarial samples that test model robustness, along with comprehensive experiments and insights into improving diagnostic accuracy using chain-of-thought reasoning and GPT-4-generated visual descriptions. On the other hand, the reviewers identified shared concerns about the dataset's balance and comprehensiveness, particularly the overrepresentation of chest X-rays, the focus on binary closed-ended questions, and insufficient validation of adversarial samples by medical experts (6yHr, vP6d, ivmY). Potential data contamination from publicly available sources raised fairness concerns, and the inclusion of clinically trivial questions reduced the dataset’s practical relevance (vP6d, 9nkQ). Additionally, unclear presentation of results, such as the distinction between clean and adversarial data, and the lack of multiple-choice questions were noted as missed opportunities (ivmY). Collectively, these issues limit the dataset's robustness, fairness, and clinical applicability.


During the discussion, the authors emphasized dataset balance with binary answers across twelve [organ, modality] combinations and defended binary closed-ended questions for their scalability and robustness, though they acknowledged the lack of open-ended tasks as a limitation. Expert validation showed high accuracy but was limited in scale, and potential data contamination concerns were addressed with proposed mitigation measures.  The reviewers acknowledged some clarifications but shared concerns about the dataset's limited focus on binary yes/no questions, which 9nkQ and ivmY found restrictive and lacking clinical relevance. vP6d remained unconvinced about data contamination mitigation, noting potential biases favoring pre-trained models, while 6yHr criticized the absence of open-ended tasks as impractical for clinical use. While some reviewers raised the score, they also pointed out the unresolved concerns about fairness, clinical applicability, and dataset comprehensiveness.


While this paper introduces a well-curated dataset and provides insights into LLM-based Med-VQA models, several limitations reduce its impact and broader applicability. The emphasis on binary close-ended questions narrows its relevance, as these tasks are generally easier to address than open-ended ones, which are more reflective of real-world clinical scenarios. This focus undermines the significance of the work, particularly given that close-ended questions in Med-VQA are typically treated as classification problems, where traditional non-LLM-based models could potentially outperform LLM-based approaches. The absence of these traditional non-LLM-based methods in the evaluation leaves the study less comprehensive and balanced. Additionally, reviewers raised concerns about potential data contamination, the limited clinical relevance of some questions, and insufficient validation of adversarial samples, which were only partially addressed.

**Additional Comments On Reviewer Discussion:**

The reviewers identified shared concerns about the dataset's balance and comprehensiveness, particularly the overrepresentation of chest X-rays, the focus on binary closed-ended questions, and insufficient validation of adversarial samples by medical experts (6yHr, vP6d, ivmY). Potential data contamination from publicly available sources raised fairness concerns, and the inclusion of clinically trivial questions reduced the dataset’s practical relevance (vP6d, 9nkQ). Additionally, unclear presentation of results, such as the distinction between clean and adversarial data, and the lack of multiple-choice questions were noted as missed opportunities (ivmY). Collectively, these issues limit the dataset's robustness, fairness, and clinical applicability.


During the discussion, the authors emphasized dataset balance with binary answers across twelve [organ, modality] combinations and defended binary closed-ended questions for their scalability and robustness, though they acknowledged the lack of open-ended tasks as a limitation. Expert validation showed high accuracy but was limited in scale, and potential data contamination concerns were addressed with proposed mitigation measures.  The reviewers acknowledged some clarifications but shared concerns about the dataset's limited focus on binary yes/no questions, which 9nkQ and ivmY found restrictive and lacking clinical relevance. vP6d remained unconvinced about data contamination mitigation, noting potential biases favoring pre-trained models, while 6yHr criticized the absence of open-ended tasks as impractical for clinical use. While some reviewers raised the score, they also pointed out the unresolved concerns about fairness, clinical applicability, and dataset comprehensiveness.

---

### Decision · Program_Chairs · 2025-01-22

Reject